# Identification of Partially Observed Linear Causal Models: Graphical Conditions for the Non-Gaussian and Heterogeneous Cases

**Jeffrey Adams**[1]*, **Niels Richard Hansen**[1], **Kun Zhang**[2]

[1]Department of Mathematical Sciences, University of Copenhagen, Denmark
[2]Department of Philosophy, Carnegie Mellon University, Pittsburgh, USA
ja@math.ku.dk, niels.r.hansen@math.ku.dk, kunz1@cmu.edu

## Abstract

In causal discovery, linear non-Gaussian acyclic models (LiNGAMs) have been studied extensively. While the causally sufficient case is well understood, in many real applications the observed variables are not causally related. Rather, they are generated by latent variables, such as confounders and mediators, which may themselves be causally related. Existing results on the identification of the causal structure among the latent variables often require very strong graphical assumptions. In this paper, we consider partially observed linear models with either non-Gaussian or heterogeneous errors. In that case we give two graphical conditions which are necessary for identification of the causal structure. These conditions are closely related to sparsity of the causal edges. Together with one additional condition on the coefficients, which holds generically for any graph, the two graphical conditions are also sufficient for identifiability. These new conditions can be satisfied even when the number of latent variables is very large. We demonstrate the validity of our results on synthetic data.

## 1 Introduction

In the standard causal discovery problem, we are given non-experimental data and aim to learn the direct causal relations between the observed variables [1, 2]. But in many applications, we do not believe that all causal variables relevant to the observed system have been measured. While some of the observed variables may interact directly, others might interact indirectly via latent mediators, and still others could be generated by latent common causes; indeed, any pair of observed variables may stand in all three relations at once. Further, the relevant latent variables may be causally related themselves. For example, responses to psychometric questionnaires are usually thought of as noisy views of various traits, and the researcher is predominately interested in the causal relations between these hidden traits and their hierarchical structure. Similarly, in financial markets, stock returns may be causally related, but may also be confounded or mediated by a complicated network of unmeasured economic and political factors.

It is natural to ask what conditions are both necessary and sufficient for the identification of such partially observed causal structures from observational data. Various sufficient conditions have been proposed; however, these conditions are rather restrictive, and are not in general necessary for identification of the full causal structure.

---

*The work presented in this article was started while JA was at CMU.

35th Conference on Neural Information Processing Systems (NeurIPS 2021).

In this work, we consider the case of linear causal models in which the overcomplete mixing matrix from the noise terms to the measured variables is identifiable up to permutation and scaling of columns. This is possible, for example, in the case of independent non-Gaussian noise [3], or when given access to heterogeneous domains in which the variances of the noise terms change independently across domains but the causal graph and weights remain constant (see Theorem 1 of our paper). We provide necessary and sufficient conditions under which the latent causal structure can be uniquely identified up to trivial indeterminicies.

## 2 Problem setup

Suppose some causal variables $\mathcal{V} = \{V_1, \ldots, V_p\}$ follow a linear structural equation model (SEM)

$$\mathbf{V} = \mathbf{F}\mathbf{V} + \varepsilon, \tag{1}$$

where $\mathbf{V} := (V_1, \ldots, V_p)^T$ is a vector of causal variables, $\mathbf{F}$ is a causal adjacency matrix that can be permuted (by simultaneous row and column permutations) to strictly lower-triangular form, and $\varepsilon = (\varepsilon_1, \ldots, \varepsilon_p)^T$ is a vector of independent noise variables. In this paper we consider two settings for $\varepsilon_i$: 1) all $\varepsilon_i$ are mutually independent and non-Gaussian; or 2) there are multiple domains, $\varepsilon_i$ are uncorrelated within each domain, and their variances change independently across domains. We will make the second assumption technically precise Section 3.

We seek necessary and jointly sufficient conditions for identifiability of $\mathbf{F}$ (up to trivial indeterminacies) in the case where only some subset of $\mathcal{V}$ (which we call $\mathcal{X}$) is measured. Thus $\mathbf{F}$ may encode observed-observed interactions, latent confounding, latent-latent interactions, and latent mediation or intermediate confounding. Our results identify $\mathbf{F}$ from the equivalence class $\mathcal{M}$, as defined in Section 2.2, of mixing matrices induced by (1). This equivalence class $\mathcal{M}$ is identifiable if, for example, the errors are non-Gaussian or if their distribution changes over time or between domains.

### 2.1 Notation

For any matrix $\mathbf{A}$ and index sets $J$ and $K$, we write $\mathbf{A}_K^J$ to denote the submatrix of $\mathbf{A}$ with columns indexed by $J$ and rows indexed by $K$. Observe that $\mathbf{A}_K^J = \mathbf{I}_K \mathbf{A} \mathbf{I}^J$. Thus, $\mathbf{F}_K^J$ describes the direct effect of $\{V_j : j \in J\}$ on $\{V_k : k \in K\}$. (Remember: causes are *up*-stream of their effects.)

The graph induced by (1) has edges $V_i \rightarrow V_j$ whenever $\mathbf{F}_j^i \neq 0$. We write $\mathrm{Pa}(V_i) := \{V_j : V_i \leftarrow V_j\}$ and $\mathrm{Ch}(V_i) := \{V_j : V_i \rightarrow V_j\}$, respectively, to denote the parents and children of $V_i$. We say that $(V_1, ..., V_k)$ constitutes a **directed path** from $V_1$ to $V_k$ if $V_i \rightarrow V_{i+1}$ for every $i \in \{1, ..., k-1\}$. Trivially, for every $V_i$, $(V_i)$ is a directed path from $V_i$ to itself; we say that a directed graph is **acyclic** (a DAG) if $(V_i)$ is the only such path. We write $\mathrm{Anc}(V_i) := \{V_j \in \mathcal{V} : V_j \text{ has a directed path to } V_i\}$ and $\mathrm{Desc}(V_i) := \{V_j : V_i \text{ has a directed path to } V_j\}$, respectively, to denote the ancestors and descendants of $V_i$. For DAGs, notice that $\mathrm{Anc}(V_i) \cap \mathrm{Desc}(V_i) = \{V_i\}$, but $V_i \notin \mathrm{Pa}(V_i) \cup \mathrm{Ch}(V_i)$.

We assume that only some subset $\mathcal{X} \subseteq \mathcal{V}$ is observed, with the remaining $\mathcal{L} = \mathcal{V} - \mathcal{X}$ being latent. We use $V_i$ to denote a generic variable, observed or latent, while $X_i \in \mathcal{X}$ denotes an observed variable and $L_j \in \mathcal{L}$ denotes a latent variable. When it is clear from context, we occasionally suppress the distinction between a variable $V_i$ and its index $i$.

### 2.2 Identification and minimality

Since $\mathbf{F}$ induces a DAG, we can always solve (1) to express the causal variables in terms of the independent noise terms:

$$\mathbf{V} = \mathbf{M}\varepsilon, \tag{2}$$

where

$$\mathbf{M} := (\mathbf{I} - \mathbf{F})^{-1} \tag{3}$$

is the mixing matrix with $\mathbf{M}_j^i$ being the net effect of $\varepsilon_i$ on $V_j$. This net effect is calculated by multiplying causal weights along paths and summing across paths. Notice that if $\mathbf{M}_j^i \neq 0$, then $V_j \in \mathrm{Desc}(V_i)$.

Because $\mathbf{L}$ is hidden, let us explicitly write $\mathbf{X}$ in terms of $\varepsilon$:

$$\mathbf{X} = \mathbf{M}_{\mathcal{X}}\varepsilon. \tag{4}$$

In both the non-Gaussian and heterogeneous settings we consider in this paper, $\mathbf{M}_\mathcal{X}$ is identifiable up to permutation and scaling of columns; that is, we can identify the equivalence class

$$\mathcal{M} = \{\mathbf{M}_\mathcal{X}\mathbf{DP} : \mathbf{DP} \in \mathcal{DP}_p\}, \tag{5}$$

where

$$\mathcal{DP}_p := \{\mathbf{DP} \in \mathbb{R}^{p \times p} : \mathbf{D} \text{ is full rank diagonal and } \mathbf{P} \text{ is a permutation matrix}\}.$$

We argue this for both settings individually in Section 3.

We say that an adjacency matrix $\mathbf{F}$ **generates** $\mathcal{M}$ if $(\mathbf{I} - \mathbf{F})_\mathcal{X}^{-1} \in \mathcal{M}$. Of course, in partially observed systems, the adjacency matrix that generates $\mathcal{M}$ is not unique. However, some of these matrices are sparser than others. In causal discovery, as in model selection more broadly, we tend to prefer the "simplest" model that adequately fits the data [4, 5]. As a result, without prior knowledge, a partially observed linear causal model cannot be identified if the population distribution can be written in terms of an equally sparse or sparser model; after all, we would never select a complicated model if a simpler model fits just as well. It is therefore natural to recast the question of identifiability to a question of maximal sparsity.

Let the $\ell_0$ "norm" of a matrix $\|\cdot\|_0$ denote the number of non-zero entries in that matrix. Then we say that a causal adjacency matrix $\mathbf{F}$ is **minimal** with respect to $\mathcal{M}$ if $\mathbf{F}$ generates $\mathcal{M}$ and $\|\hat{\mathbf{F}}\|_0 \geq \|\mathbf{F}\|_0$ for any $\hat{\mathbf{F}} \neq \mathbf{F}$ that generates $\mathcal{M}$.

Let $\mathcal{F}$ denote the class of minimal adjacency matrices that generate $\mathcal{M}$. Clearly, since $\mathcal{M}$ is identifiable, so is $\mathcal{F}$. We say that an adjacency matrix $\mathbf{F}$ is **identified up to trivialities** if

$$\mathcal{F} = \left\{(\mathbf{DP})^{-1}\mathbf{FDP} : \mathbf{DP} \in \mathcal{DP}_p \text{ with } (\mathbf{DP})_\mathcal{X}^\mathcal{X} = \mathbf{I}\right\}. \tag{6}$$

The only indeterminacy remaining in $\mathcal{F}$ amounts to re-indexing and re-scaling the latent factors.

A word of caution is in order. Because the adjacency matrix that generates $\mathcal{M}$ is not unique in the partially observed case, it is only possible to talk about identification with respect to some selection principle. Throughout this work we use minimality as such a selection principle – indeed we define identification in terms of it. As justification, in Section 5.1, we describe one class of non-minimal adjacency matrices which are pathological and whose exclusion is desirable; further, in Section 6, we show that existing works make assumptions even stronger than minimality; further still, in Section 7, we show that popular model selection criteria like BIC favor minimal graphs. Nevertheless, just as BIC is not always the most sensible criterion for model selection, so minimality is not always the most sensible principle for an identification theory. For example, Figure 1 shows a non-minimal graph which is not pathological. Thus, if a practitioner believes the true partially observed causal model to be non-minimal, they should content themselves with partial identification (c.f. [6]).

In Sections 4 and 5, we express identification up to trivialities in terms of two local graphical conditions, which are much easier to check than (6). But first, we return to the identifiability of $\mathcal{M}$.

## 3 Sufficient conditions for identification of $\mathcal{M}$

The main results of Sections 4 and 5 rely on the identifiability of $\mathcal{M}$, which is theoretically guaranteed in the two settings we consider in this paper. In the first setting, $\varepsilon_i$ are assumed to be independent and non-Gaussian. Then according to Theorem 3 by Eriksson and Koivunen [7], $\mathcal{M}$ is identifiable from the distribution of $\mathbf{X}$. The task of estimating $\mathcal{M}$ from $\mathbf{X}$ is known as Overcomplete Independent Component Analysis (OICA) [3], and in practice this task is known to be computationally difficult [8].

In the second setting, $\varepsilon_i$ are uncorrelated from each other with changing variances across multiple domains (or over time) and $\mathbf{M}_\mathcal{X}$ has full row rank (which is always the case for acyclic models). Note that in this setting, while the components of $\varepsilon$ are mutually independent within each domain, they are not necessarily mutually independent across domains because their variances may be dependent across domains. This setting is expected to apply to a number of nonstationary scenarios including brain signal analysis, and the following theorem establishes the corresponding identifibiality of $\mathcal{M}$. Besides complementing Theorem 3 of Eriksson and Koivunen [7] as an alternative foundation for our identification work, the identifiability of $\mathcal{M}$ in this setting may be of independent interest in the fields of blind source separation and system identification.

**Theorem 1.** *Suppose we have observed* $\mathbf{X}$ *generated according to the mixing procedure (4) in a number of domains,* $t = 1, 2, ..., T$, *where* $\mathbf{M}_{\mathcal{X}}$ *has full row rank. Assume that* $\varepsilon_i$ *are uncorrelated in each domain and that their variances in domain t, denoted by* $\sigma_{ti}^2$, *change independently across domains in the sense that* $\mathbf{S}$, *whose* $(t, i)$*-th entry is* $\sigma_{ti}^2$, *has full column rank. Further assume that each* $|\mathcal{X}|$ *columns of* $\mathbf{M}_{\mathcal{X}}$ *are linearly independent and that* $p \leq 2|\mathcal{X}| - 2$. *Then if* $\mathbf{X}$ *admits a model* $\mathbf{X} = \tilde{\mathbf{M}}_{\mathcal{X}}\tilde{\varepsilon}$, *where* $\tilde{\varepsilon}$ *also follows the above assumption on* $\varepsilon$, *then every column of* $\tilde{\mathbf{M}}_{\mathcal{X}}$ *must be proportional to a column of* $\mathbf{M}_{\mathcal{X}}$ *and vice versa.*

Note that this theorem gives sufficient conditions for the identifiability of $\mathcal{M}$; our empirical results suggest that they are not necessary.

## 4 Necessary conditions for identification of F up to trivialities

In this section, we introduce our identification conditions and show that they are necessary for $\mathbf{F}$ to be identified up to trivialities. The identification conditions are graphical conditions described in terms of "bottlenecks" and "redundancies."

Let $J$, $K$, and $B$ be subsets of the nodes of a directed graph. Note that they need not be mutually disjoint. We say that $B$ is a **bottleneck** from $J$ to $K$ if, for every $j \in J$ and every $k \in K$, each directed path from $j$ to $k$ includes some $b \in B$. A bottleneck $B$ from $J$ to $K$ will be called **minimal** if every bottleneck $B'$ from $J$ to $K$ has $|B'| \geq |B|$, and **unique minimal** if the inequality is strict for $B' \neq B$. Note that bottlenecks do not in general $d$-separate $J$ and $K$ along all paths, but only directed paths from $J$ to $K$.

It is clear from the definition that, for each $V_i$, $\mathrm{Ch}(V_i)$ is a bottleneck from $\mathrm{Ch}(V_i)$ to $X$. However, for identification, we further require:

**Condition 1** (Bottleneck). *For every* $V_i$, $\mathrm{Ch}(V_i)$ *is the unique minimal bottleneck from* $\mathrm{Ch}(V_i)$ *to* $\mathcal{X}$.

As illustrated in Figure 1, the bottleneck condition ensures that if we try to "explain" the net effect of $V_i$ on $\mathcal{X}$ by replacing $\mathrm{Ch}(V_i)$ with any subset of $\mathrm{Desc}(V_i)$, the result is a denser graph. As illustrated in Figure 3, the strong non-redundancy condition will further ensure that we cannot "explain" the effect of $V_i$ on $\mathrm{Ch}(V_i)$ via some of its *non*-descendants:

**Condition 2** (Strong Non-Redundancy). *For all* $L_i, V_j$, *if* $\mathrm{Ch}(L_i) \subseteq \mathrm{Ch}(V_j) \cup \{V_j\}$ *then* $L_i = V_j$.

Figure 2 shows a graph that satisfies both of these conditions. To build intuition, let us list some simple consequences of these conditions. By the bottleneck condition, each variable must have fewer than $|\mathcal{X}|$ children; but if a variable has no latent children, then the bottleneck condition is satisfied trivially for that variable. By strong non-redundancy, each latent variable must have at least two children. For any pair $(L_i, V_j)$, if $L_i$ is an ancestor but not a parent of $V_j$, or has more than one directed path to $V_j$, then strong non-redundancy is satisfied for that pair. If $V_j$ is a parent of $L_i$ and they violate strong non-redundancy, then the bottleneck condition is violated for $V_j$.

**Theorem 2.** *If* $\mathbf{F}$ *is identified up to trivialities, then the graph induced by* $\mathbf{F}$ *satisfies the bottleneck and strong non-redundancy conditions.*

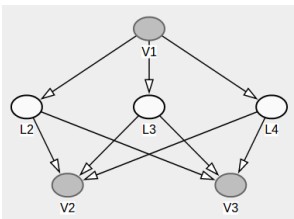 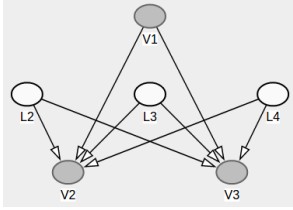 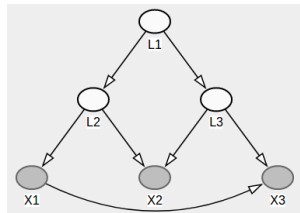

Figure 1: An egregious violation of the bottleneck condition. Left: $\{V_2, V_3\}$ is a strictly smaller bottleneck from $\mathrm{Ch}(V_1)$ to $\mathcal{X}$. Right: a sparser yet observationally equivalent graph. Although both graphs also violate strong non-redundancy, egregious bottleneck violations are not always redundant.

Figure 2: A simple graph illustrating our structural conditions. $L_1$ satisfies the bottleneck condition. $L_2$ and $L_3$ are non-redundant as each has a child the other does not. $X_1$ and $L_3$ are non-redundant as $L_3 \to X_2$ and $X_1 \notin \mathcal{L}$.

Thus the bottleneck and strong non-redundancy conditions are necessary for identification of $\mathbf{F}$ up to trivialities. In Section 5, we further show that they (along with a very mild constraint on the causal weights) are also jointly sufficient conditions.

If $\mathrm{Ch}(V_i)$ is not at least a *minimal* bottleneck for every $V_i$, then $\mathbf{F} \notin \mathcal{F}$. Figure 1 shows one example of such a violation of the bottleneck condition. Otherwise, as long as bottleneck faithfulness is also satisfied, $\mathcal{F}$ is an equivalence class of equally sparse latent structures which all violate at least one of the bottleneck and strong non-redundancy conditions. The nature of these indeterminacies is depicted in Figure 3. In Figure 2 we show a simple yet illustrative example in which both conditions are satisfied.

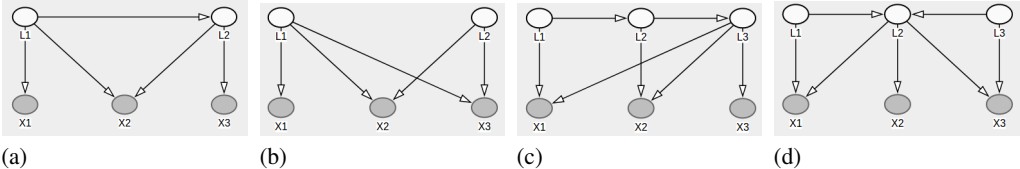

(a)  (b)  (c)  (d)

Figure 3: Two equivalence classes. (a) and (b) are equivalent, the former violating the bottleneck condition ($\mathcal{X} \neq \mathrm{Ch}(L_1)$ is a minimal bottleneck from $\mathrm{Ch}(L_1)$ to $\mathcal{X}$) and the latter strong non-redundancy ($\mathrm{Ch}(L_2) \subseteq \mathrm{Ch}(L_1)$). (c) and (d) are equivalent, both violating strong non-redundancy.

# 5 Sufficient conditions for identification of F up to trivialities

In the previous section, we introduced two structural conditions which must be satisfied for $\mathbf{F}$ to be identifiable up to trivialities. In this section, we prove that they (along with "bottleneck faithfulness," a very mild constraint on the causal weights) are also jointly sufficient. Throughout, we assume that $\mathbf{X}$ is generated according to (1). In particular, we assume that $\mathcal{M}$ is identifiable, for example due to Theorem 3 of Eriksson and Koivunen [7] or Theorem 1 of the present work.

## 5.1 Bottleneck faithfulness

First, we connect ranks of submatrices of $\mathbf{M}$ to minimal bottlenecks of its corresponding graph.

**Proposition 1.** *Let $B$ be a minimal bottleneck from $J$ to $K$. Then $\mathrm{Rank}(\mathbf{M}_K^J) \leq |B|$.*

Strict inequality in Proposition 1 for some minimal bottleneck $B$ from $J$ to $K$ can make $\mathbf{F}$ non-identifiable – even if the bottleneck condition and strong non-redundancy hold. For instance, both graphical conditions hold for

$$\mathbf{X} = \begin{bmatrix} 0 & 1 & -1 \\ 2 & 2 & 0 \\ 3 & 3 & 0 \\ 4 & 0 & 4 \end{bmatrix} \mathbf{L} + \varepsilon_X, \quad \mathbf{L} = \varepsilon_L,$$

but $\mathrm{Rank}(\mathbf{M}_{\mathcal{X}}^{\mathcal{L}}) = 2$ while the minimal bottleneck from $\mathcal{L}$ to $\mathcal{X}$ is $\mathcal{L}$ with $|\mathcal{L}| = 3$. The system

$$\mathbf{X} = \begin{bmatrix} 0 & 1 & -1 \\ 0 & 2 & 0 \\ 0 & 3 & 0 \\ 0 & 0 & 4 \end{bmatrix} \mathbf{L} + \varepsilon_X, \quad \mathbf{L} = \begin{bmatrix} 0 & 0 & 0 \\ 1 & 0 & 0 \\ 1 & 0 & 0 \end{bmatrix} \mathbf{L} + \varepsilon_L$$

generates the same mixing matrix, $\mathbf{M}_{\mathcal{X}}$, but has a strictly sparser graph. Thus to ensure identifiability, we assume that the causal coefficients satisfy:

**Condition 3** (Bottleneck Faithfulness). *For every $J \subseteq \mathcal{V}, K \subseteq \mathcal{X}$, if $B$ is a minimal bottleneck from $J$ to $K$, then $\mathrm{Rank}\left(\mathbf{M}_K^J\right) = |B|$.*

In the supplementary material we characterize the set of adjacency matrices $\mathbf{F}$ that are bottleneck faithful for a given graph. In particular, we show that a generic $\mathbf{F}$ is bottleneck faithful.

Interestingly, in linear systems, classical faithfulness is a special case of bottleneck faithfulness. $\mathrm{Rank}(\mathbf{M}_K^J) = 0$ is a violation of classical faithfulness if there is a minimal bottleneck $B \neq \varnothing$ from

$J$ to $K$. That is, if there is a path from $J$ to $K$ but the path coefficients cancel out so that the net effect of $J$ on $K$ is 0, the system is not faithful to the graph. Bottleneck faithfulness generalizes this so that the net effect of $J$ on $K$ must have maximal rank for the given graph.

## 5.2 Identifiability

In this subsection, we show that if the bottleneck condition, strong non-redundancy, and bottleneck faithfulness hold for $\mathbf{F}$, then $\mathbf{F}$ is identifiable up to trivialities. Throughout, we assume the three conditions hold.

Our approach is illustrated by the following computation. For any $V_i$,

$$\mathbf{M}_{\mathcal{X}}(\mathbf{I} - \mathbf{F})^i = \mathbf{I}_{\mathcal{X}}^i. \tag{7}$$

Let $J = \mathrm{Ch}(V_i)$. Since the support of $\mathbf{F}^i$ is $J$, the equation

$$(\mathbf{M} - \mathbf{I})_{\mathcal{X}}^i = \mathbf{M}_{\mathcal{X}}^J \mathbf{x}. \tag{8}$$

always has a solution at $\mathbf{x} = \mathbf{F}_J^i$. In fact, (under the three assumptions of this section) this solution is unique:

**Lemma 1.** *Let $J = \mathrm{Ch}(V_i)$. Then the unique solution to (8) is given by $\mathbf{x} = \mathbf{F}_{\mathrm{Ch}(V_i)}^i$.*

But there is a version of (8) for each $J \subseteq \mathcal{V}$. For which other choices of $J$ does (8) have a solution? Clearly a solution always exists if $J \supseteq \mathrm{Ch}(V_i)$. On the other hand, we can guarantee that a solution with $|J| \leq |\mathrm{Ch}(V_i)|$ is only possible if $J$ contains an ancestor of $V_i$:

**Lemma 2.** *Suppose $J \subseteq \mathcal{V} - \mathrm{Anc}(V_i)$. If $(\mathbf{M} - \mathbf{I})_{\mathcal{X}}^i \in \mathrm{Range}\left(\mathbf{M}_{\mathcal{X}}^J\right)$, then $|J| \geq |\mathrm{Ch}(V_i)|$, with equality if and only if $J = \mathrm{Ch}(V_i)$.*

By Lemma 2, if any superset of $\mathrm{Ch}(V_i)$ containing no ancestors of $V_i$ is identifiable, then $\mathrm{Ch}(V_i)$ is also identifiable. Next, we will show how such a superset of $\mathrm{Ch}(V_i)$ can be identified.

Let $\mathcal{V}_k \subseteq \mathcal{V}$ denote the variables whose longest path to $\mathcal{X}$ has fewer than $k$ nodes. More formally, we define recursively

$$\mathcal{V}_0 := \varnothing, \tag{9}$$
$$\mathcal{V}_{k+1} := \{V_i \in \mathcal{V} : \mathrm{Ch}(V_i) \subseteq \mathcal{V}_k\}, \quad \text{for } k \geq 0. \tag{10}$$

Naturally, we define $\mathcal{X}_k := \mathcal{V}_k \cap \mathcal{X}$ and $\mathcal{L}_k := \mathcal{V}_k \cap \mathcal{L}$. Notice that $\mathcal{V}_k$ is strictly increasing, and is induced by the topological ordering on $\mathcal{V}$.

**Proposition 2.** *For all $k$, either $\mathcal{V}_k \subset \mathcal{V}_{k+1}$, or $\mathcal{V}_k = \mathcal{V}$.*

Further, for each $k \geq 0$, define

$$\mathcal{J}_{k+1}(V_i) := \underset{J \in \left\{J \subseteq \mathcal{V}_k : \mathbf{M}_{\mathcal{X}_k}^i \in \mathrm{Range}\left(\mathbf{M}_{\mathcal{X}_k}^J\right)\right\}}{\arg\min} |J|. \tag{11}$$

Intuitively, this denotes the set of minimal choices for $J \subset \mathcal{V}_k$ such that (8) has a solution. From Lemma 2, we know that $\mathcal{J}_{k+1}(V_i) = \{\mathrm{Ch}(V_i)\}$ if $V_i \in \mathcal{V}_{k+1} - \mathcal{V}_k$. The construction of (11) allows us to generalize Lemma 2 to describe which versions of (8) have solutions when we are not sure of the causal order.

**Lemma 3.** *For every $k \geq 0$, let $\mathcal{V}_k$ and $\mathcal{J}_k(V_i)$ be defined as above. Then $V_i \in \mathcal{V}_{k+1} - \mathcal{V}_k$ if and only if all of the following hold:*

1. *$V_i \notin \mathcal{V}_k$;*

2. *$|\mathrm{Support}(\mathbf{M}_{\mathcal{X}}^i) - \mathcal{X}_k| \leq 1$;*

3. *$|\mathcal{J}_{k+1}(V_i)| = 1$; and*

4. *for all $V_j \neq V_i$ satisfying points 1 and 2, $\mathbf{M}_{\mathcal{X}_k}^j \notin \mathrm{Range}\left(\mathbf{M}_{\mathcal{X}_k}^{\mathcal{J}_k(V_i)}\right)$.*

As a result of Lemma 3, each $\mathcal{V}_k$ is identifiable. Clearly, if $V_i \in \mathcal{V}_{k+1}$, then $\mathrm{Ch}(V_i) \in \mathcal{V}_k$ and $\mathcal{V}_k \cap \mathrm{Anc}(V_i) = \varnothing$. Hence by Lemma 2, $\mathrm{Ch}(V_i)$ is also identifiable. Thus the full DAG is identifiable, and each column of $\mathbf{M}_{\mathcal{X}}$ can be associated with the corresponding node in the DAG. However, Lemmas 1, 2, and 3 are not enough to distinguish which nodes correspond to latent variables and which correspond to observed variables; we have yet to pair each $X_i$ with its net effects $\mathbf{M}_{\mathcal{X}}^i$. Resolving this final indeterminacy is not hard. Intuitively, the vector of $\mathbf{M}_{\mathcal{X}}$ corresponding to $X_i$ must have non-zero coefficients in the $i$-th slot while every vector corresponding to descendants of $X_i$ will not. Lemma 4 formalizes this observation.

**Lemma 4.** $X_i \in \mathcal{X}_{k+1}$ *if and only if* $X_i \in \mathcal{V}_{k+1}$ *and* $\mathrm{Support}\,(\mathbf{M}_{\mathcal{X}}^{X_i}) - \mathcal{X}_k = \{i\}$.

Together, Lemmas 1, 2, 3, and 4 imply that $\mathbf{F}$ is identifiable if $\mathbf{M}_{\mathcal{X}}$ is identifiable. Of course, we do not know $\mathbf{M}_{\mathcal{X}}$—only $\mathcal{M}$. Nevertheless, Lemmas 2, 3, and 4 do not involve the scaling and permutation of $\mathbf{M}_{\mathcal{X}}$—only the linear dependencies of its columns. Some simple calculation shows that Lemma 1 can be used to put any $\mathbf{M}_{\mathcal{X}}\mathbf{PD} \in \mathcal{M}$ in one-to-one correspondence with $(\mathbf{PD})^{-1}\mathbf{F}(\mathbf{PD})$.

**Theorem 3.** *Suppose* $\mathbf{F}$ *satisfies generalized non-redundancy, bottleneck faithfulness, and the bottleneck condition. Then* $\mathbf{F}$ *is identifiable up to indeterminacies.*

# 6 Relation to existing work

Constraint- and score-based approaches to causal discovery based on conditional independence testing—such as SGS [1], IC [2], PC [9], GES [10], and FGS [11]—generally focus on the causally sufficient case. These algorithms identify the Markov equivalence class of graphs which all encode the same set of conditional independence relations. While some methods based on conditional independence tests, such as FCI [12] and RFCI [13], are able to relax the assumption of causal sufficiency, their focus is on learning the causal relations between observed variables and distinguishing them from spurious dependencies induced by shared latent ancestors. Such methods recover only limited information about the latent structure, as only the most basic information about latent structure is identifiable from conditional independence relations alone. For one review of causal discovery methods, see Spirtes and Zhang [14].

It is possible to go beyond the equivalence class with additional assumptions on causal mechanisms [14]. In particular, linear non-Gaussian models have been studied extensively. In the causally sufficient case, Shimizu et al. [15] leverages acyclicity of the causal relations and the identifiability of the square ICA problem [16, 17] to show how the causal adjacency matrix can be identified, while Lacerda et al. [18] further estimate a subclass of cyclic causal models. In both cases, one may replace the non-Gaussian noise assumption with the heterogeneous noise assumption (in the formal sense of Theorem 1) and the identifiability results still hold [19].

By contrast, previous works on partially observed linear non-Gaussian models only study certain special cases in which the models are partially identifiable. Hoyer et al. [20] describe a procedure to convert partially observed causal models to a canonical form in which no latent variable has any parents. They further provide an algorithm which recovers all canonical forms consistent with the observed overcomplete basis $\mathcal{M}$, which is identifiable by OICA [3]. This recovered equivalence class of observationally equivalent canonical forms can be huge, and by definition can neither identify causal relations among latent confounders nor distinguish latent confounders from latent mediators.

More recently, Lemma 5 of Salehkaleybar et al. [6] states that if $\mathcal{M}$ is identifiable, then the causal order among observed variables is identifiable if classical faithfulness holds between all variables. Their condition is strictly weaker than ours; as we discuss in Section 5.1, classical faithfulness is entailed by bottleneck faithfulness and imposes no graphical conditions. That a weaker condition suffices for their task is not surprising, since their task is strictly easier than ours; if $\mathbf{F}$ is identified up to trivialities then the causal order among observed variables is also identified (while the causal order alone tells very little about $\mathbf{F}$). Lemma 5 has a reassuring consequence for our work: even if the graphical conditions for total identification fail to apply, the causal order of $\mathcal{X}$ is still identified.

If the practitioner is further interested in the observed variables' net effects on one another, $(\mathbf{M} - \mathbf{I})_{\mathcal{X}}^{\mathcal{X}}$, then additional graphical assumptions are needed. Theorem 16 of Salehkaleybar et al. [6] provides one condition sufficient for this purpose: no latent variable $L_i$ has precisely the same observed descendants as any observed variable $X_j$ (formally, for all $L_i, X_j, \mathrm{Desc}(L_i) \cap \mathcal{X} \neq \mathrm{Desc}(X_j) \cap \mathcal{X}$).

Again, this being a relatively easy subtask of the problem we consider, it is not surprising that our conditions are not strictly weaker. With that said, our conditions are not strictly stronger, either; the bottleneck and strong non-redundancy conditions can be satisfied even when $L_i$ and $X_j$ have the same observed descendants, as shown in Figure 4.

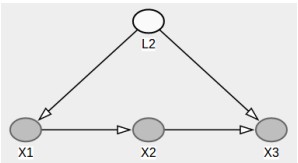 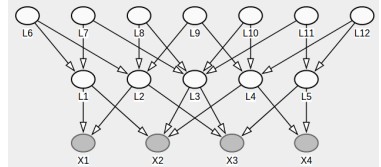 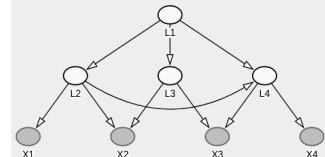

Figure 4: Examples of graphs identifiable from $\mathcal{M}$. From left to right: a graph where $\mathrm{Desc}(L) \cap \mathcal{X} = \mathrm{Desc}(X_1) \cap \mathcal{X}$; a widening hierarchical structure; a hierarchical structure with intra-layer relations.

To recover causal structures of the hidden variables, many results rely on strong assumptions about clusters of pure variables (sets of observed variables which each share a latent confounder and have no other parents). For example, Spearman's classical Tetrad condition [21] identifies latent causes with four pure observed children from covariance information alone. In the linear non-Gaussian case, existing work reduces the number of pure observed children to three [22], and more recently to two [23, 24]. Clearly, these are all special cases of both the bottleneck and non-redundancy conditions. As such, our graphical assumptions are strictly weaker.

**Proposition 3.** *Suppose each $L_i$ in a partially observed DAG has at least two pure children (latent or observed). Then the DAG satisfies the bottleneck and non-redundancy conditions.*

However, identification is possible even when no latent confounder has any pure children; for example, Anandkumar et al. [25] present a model in which latent variables with no pure children are identifiable. Rather than purity, they require a graph expansion property—for all non-singleton $S \subseteq \mathcal{L}$, $|\bigcup_{L_i \in S} \mathrm{Ch}(L_i) \cap \mathcal{X}| \geq |S| + d_{\max}$, where $d_{\max} = \max_i |\mathrm{Ch}(L_i) \cap \mathcal{X}|$ —as well as a rank condition on $\mathbf{F}_{\mathcal{X}}^{\mathcal{L}}$ which places hard-to-check graphical constraints on the model and bounds $|\mathcal{L}| \leq \frac{1}{3}|\mathcal{X}|$. Non-redundancy among latent variables can be derived from the expansion property by considering the case where $|S| = 2$, and the bottleneck condition by considering $S = \{L_i\} \cup \mathrm{Ch}(L_i) \cap \mathcal{L}$. Thus in one sense, our conditions can be seen as a refinement on the expansion property; however, we also remove the many hard-to-check graphical consequences of the rank condition, and further show that many graphs even with $|\mathcal{L}| \gg |\mathcal{X}|$ are identifiable. For example, Figure 4 shows an identifiable hierarchical model in which the number of latent variables increases with depth. Moreover, we show that our conditions are sufficient for identifying hierarchical structures in which variables in the same layer are causally related. Figure 4 shows one such system.

**Proposition 4.** *Suppose $\mathbf{F}$ satisfies the rank and graph expansion conditions of Anandkumar et al. [25]. Then $\mathbf{F}$ also satisfies the bottleneck and strong non-redundancy conditions.*

Propositions 3 and 4 show that our conditions are indeed more general than previous identification conditions; not only do our conditions allow and identify causal relations among observed variables, they also identify latent structures which no previous works could. (See, for example, Figure 4.) Furthermore, in light of Theorems 2 and 3, they show that many existing works implicitly took sparsity of causal edges as a useful primitive for what it means for a partially observed causal model to be identifiable. Such a primitive is widely used throughout causal discovery, even in the causally sufficient case [4, 5].

Although many of these conditions for latent structure identification rely on non-Gaussian independent noise, direct estimation of the mixing matrix is often avoided in practice, especially in the causally insufficient case, as estimation of the overcomplete mixing matrix is computationally challenging [8]. Estimation of the mixing matrix can be avoided by directly using the independent additive noise assumption and exploiting graphical conditions such as causal sufficiency or purity. For example, in the causally sufficient case, $\mathrm{Pa}(X_i)$ is identifiable by regressing $X_i$ on $\mathcal{Z} \subseteq \mathcal{X}$ and testing the independence of the regression residuals and $\mathcal{Z}$ [26]. This approach may be adapted to the non-linear [27, 28] and post-nonlinear [29] cases. Tashiro et al. [30] extend this idea to identify causal relations among observed variables in the causally insufficient case, and a related condition is developed by Xie et al. [24] to identify one special type of confounder. However, such methods owe their efficiency to the strong structural conditions under which they guarantee identifiability. As the bottleneck

and non-redundancy conditions are much more general, this naturally complicates the question of estimation.

# 7 Estimation

In Theorems 1 and 3, we have shown that a causal system which satisfies the conditions of Section 4 is uniquely identifiable whenever $\mathbf{M}_{\mathcal{X}}$ is identifiable. However, as indicated in Section 6, estimation of $\mathbf{M}_{\mathcal{X}}$ from homogeneous non-Gaussian data—for example, by overcomplete ICA—is computationally hard. Further, whereas the estimation algorithms presented in [15], [18], and [6] require the practitioner to test which entries of $\mathbf{M}_{\mathcal{X}}$ are exact zeros, a naive algorithm inspired by Lemmas 1, 2, 3, and 4 would further require them to test which submatrices' singular values are exact zeros. Such an algorithm is not advisable.

As a proof of concept, we therefore focus our experiments on partially observed linear causal models in the heterogeneous case. In this setting, $\mathbf{F}$ can be learned directly by optimizing the regularized likelihood with respect to $\mathbf{F}$, given the sample covariance matrices of $\mathbf{X}$. We leave more efficient estimation in more general settings to future work.

## 7.1 Simulations

Suppose we have access to samples from $T$ heterogeneous domains. The data in the $t$-th domain follow

$$\mathbf{V} = \mathbf{F}\mathbf{V} + \varepsilon, \tag{12}$$

where $\varepsilon \sim \mathcal{N}(0, \Sigma_t)$ for diagonal $\Sigma_t$. Then in the $t$-th domain,

$$\mathbf{X} = \mathbf{M}_{\mathcal{X}}\Sigma_t\mathbf{M}_{\mathcal{X}}^T. \tag{13}$$

The negative log likelihood is

$$-2\ell\ell(\mathbf{F}, \Sigma) = \sum_{t=1}^{T} n_t \left( |\mathcal{X}| \log(2\pi) + \log\det(\mathbf{S}_t) + \mathrm{Tr}\left( \mathbf{S}_t^{-1}\hat{\mathbb{E}}\left[ \mathbf{x}_t\mathbf{x}_t^T \right] \right) \right), \tag{14}$$

where $\mathbf{x}_{t,i}$ is the $i$-th row of the design matrix for the $t$-th domain, $\hat{\mathbb{E}}\left[ \mathbf{x}_t\mathbf{x}_t^T \right]$ is the empirical second moment of the $d$-th domain, $\mathbf{S}_t = \mathbf{M}_{\mathcal{X}}\Sigma_t\mathbf{M}_{\mathcal{X}}^T$, and $n_t$ is the sample size in the $t$-th domain. If $\mathbf{X}$ is generated according to (12), the independent change condition in Theorem 1 holds for the noise variances, and $\mathbf{F}$ satisfies the assumptions of Section 4, then by Theorems 1 and 3, $\mathbf{F}$ is identifiable up to trivialities. Hence we can in principle optimize the regularized log likelihood.

As a sanity check for our theoretical results, we simulate data according to (12); for every identifiable graph structure with three observed variables and at most five directed edges, we generated ten causal adjacency matrices with weights randomly drawn from $(-0.9, -0.5) \cup (0.5, 0.9)$. We estimate $\mathbf{F}$ and $|\mathcal{L}|$ by minimizing the BIC via exhaustive search. By enumerating candidate graphs from sparsest to densest, a single search could take anywhere from 10 to 60 minutes on an Intel core i7 processor.

To verify that our estimation method was actually leveraging the noise's heterogeneity, we ran the experiment with one domain and 5000 observations. Only 3% of graphs were identified. Not surprisingly, only 15% of learned graphs had any latent variables at all. Increasing the number of domains from 1 to 3 but keeping the total sample size at 5000 (i.e. 1666 per domain) improved the rate of structure identification to 50% of trials.

With 5 domains and 500 samples per domain, the correct graph is identified on 50% of trials; with 1000 samples per domain, this improves to 70%; and with 10 000, this further improves to 80%. In every case that the wrong graph was recovered, the equivalence class of mixing matrices $\hat{\mathcal{M}}$ generated by $\hat{\mathbf{F}}$ had incorrect support, perhaps due to insufficient domains or accidentally coupled changes in the noise variances. This supports the main theory of Theorem 3, which, in light of Theorem 1, claims that the structure of $\mathbf{F}$ is uniquely determined *from a correctly identified* $\mathcal{M}$. We report detailed results in the supplement for all graphs studied.

To verify our claims in Theorem 2, we also tested simulated data from ten non-identifiable partially observed DAGs—six of which stand in the main equivalence class relations of Figure 3, and

four of which are not minimal. Not surprisingly, members of the same equivalence class were indistinguishable, each system achieving the same log likelihood up to eight significant digits.

Because exhaustive search over graphs is computationally expensive (even for this toy problem, there are 1759 graphs, and so 1759 non-convex optimization problems), it would be desirable to instead optimize the L1-penalized negative log likelihood:

$$\mathrm{L}(\mathbf{F}, \Sigma) = -2\ell\ell(\mathbf{F}, \Sigma) + \lambda \sum |F_{i,j}|. \tag{15}$$

As before, we simulated data from (12). Numerical experiments verify that $\mathrm{L}(\mathbf{F}^0, \Sigma^0)$ is very near a local minimum for all practical $\lambda > 0$, where $(\mathbf{F}^0, \Sigma^0)$ denotes the true adjacency matrix and noise covariances. However, experiments also suggest that this local minimum is generally quite far from the global minimum, both in parameter space and in L1 loss. Moreover, while the L1 penalty successfully drives many parameters to zero (as we would expect), our experiments frequently converge to minima which are denser than the true system. Intuitively, the L1 penalty does not care about the density of $\hat{\mathbf{F}}$; a dense system with small coefficients may have a comparable L1 penalty as a sparse system with large coefficients. Naturally, denser systems are better equipped to fit the observed domain covariances. We summarize these experiments in the supplement.

## 8   Conclusion and discussions

In many fields, we do not believe that all causally relevant variables have been measured. In such partially observed settings, beyond accurately estimating the causal relations among observed variables, practitioners may want to further identify the causal relations among the hidden variables which generate the observed data. Inspired by this issue, we have contributed to the identification theory of partially observed linear causal models by providing necessary and sufficient graphical conditions for the identification of the full causal graph. Throughout, we assume the additive noise terms in the structural equation model follow non-Gaussian distributions or have independently changing variances across time or between domains. Such assumptions, unlike the single-domain Gaussianity assumption, render the mixing procedure from the noise terms to the observed variables identifiable up to the permutation and scaling of columns, thereby facilitating our final identifiability results. These conditions are expected to be applicable to a wide variety of partially observed structures. To deal with real applications, efficient estimation methods are needed, and we hope our theoretical identifiability results will stimulate algorithmic development to finally solve this important causal discovery problem. As future work, we will focus on developing practical estimation methods and extending our results to nonlinear cases.

## Acknowledgements

This project has received funding from the European Union's Horizon 2020 research and innovation programme under the Marie Skłodowska-Curie grant agreement No 801199. 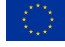

The work presented in this article was supported in part by Novo Nordisk Foundation Grant NNF20OC0062897.

This work was supported in part by the National Institutes of Health (NIH) under Contract R01HL159805, by the NSF-Convergence Accelerator Track-D award #2134901, by the United States Air Force under Contract No. FA8650-17-C7715, and by a grant from Apple Inc. The NIH or NSF is not responsible for the views reported in this article.

We are grateful to the anonymous reviewers for their careful reading and helpful comments.

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
