# Supplementary Material: Identification of Partially Observed Linear Causal Models

**Jeffrey Adams**[1], **Niels Richard Hansen**[1], **Kun Zhang**[2]

[1]Department of Mathematical Sciences, University of Copenhagen, Denmark
[2]Department of Philosophy, Carnegie Mellon University, Pittsburgh, USA
ja@math.ku.dk,    niels.r.hansen@math.ku.dk,    kunz1@cmu.edu

## 1   Proofs and more details

For convenience, we reserve the matrix $\mathbf{R}_B^J$ to denote the net effect of $J$ on $B$ *prior* to any mixing effects among $B$—that is, on the subgraph where $\mathrm{Ch}(V_b)$ has been set to $\varnothing$ for each $V_b \in B$.

**Proposition 3.** *For any $J, B \subseteq \mathcal{V}$, $\mathbf{R}_B^J = \left(\mathbf{M}_B^B\right)^{-1}\mathbf{M}_B^J$. Moreover, if $B$ is a bottleneck from $J$ to $K$, then $\mathbf{M}_K^J = \mathbf{M}_K^B \mathbf{R}_B^J$.*

### 1.1   Theorem 1 and its proof

Let us present the complete theorem first, and then give its proof. Let $n$ be the dimensionality of $\mathbf{X}$. Remeber $p$ is the number of noise terms. In the case where $n = p$, $\mathbf{M}_{\mathcal{X}}$ in (4) is a square matrix, and its identifiability from $\mathbf{X}$ up to column rescaling and permutations has been provided by Matsuoka et al. [1], but we are concerned with the case where $n < p$. In the case where the noise terms $\varepsilon_i$ are non-Gaussian, the identifiability of $\mathbf{M}_{\mathcal{X}}$ up to column rescaling and permutations was also given in the literature [2], inspired by the results in [3]. Although the corresponding OICA problem may be difficult to solve in practice, this identifiability result is nice in that it holds true even if $p$ is much larger than $n$. The heterogeneous variance case seems complementary: its maximum likelihood estimation procedure is simple, but our proof of it uses a constraint on $p$, given a fixed $n$ (this condition is sufficient, but may be unnecessary, as illustrated by our simulation results), as given in the following theorem.

Before presenting Theorem 1, let us give the following lemma, which will be needed in the proof of Theorem 1.

**Lemma 5.** *Suppose matrix $\mathbf{K} \in \mathbb{R}^{n \times n}$ has linearly independent columns, i.e., $Rank(\mathbf{K}) = n$. Let $\mathring{\mathbf{K}} = \mathbf{K} - d \cdot \mathbf{1}^{\mathsf{T}}$, where $d \in \mathbb{R}^n$ and $\mathbf{1}$ is the length-$n$ vector of all 1's. Then for any $d$, $Rank(\mathring{\mathbf{K}}) \geq n - 1$.*

*Proof.* Since $\mathbf{K}$ in invertible, let $f := \mathbf{K}^{-1} \cdot d$. Then $\mathring{\mathbf{K}} = \mathbf{K} - d \cdot \mathbf{1}^{\mathsf{T}} = \mathbf{K}(\mathbf{I} - f \cdot \mathbf{1}^{\mathsf{T}})$, where $\mathbf{I}$ denotes the identity matrix. Since $\mathbf{K}$ has full rank, $\mathrm{Rank}(\mathring{\mathbf{K}}) = \mathrm{Rank}(\mathbf{K}(\mathbf{I} - f \cdot \mathbf{1}^{\mathsf{T}})) = \mathrm{Rank}(\mathbf{I} - f \cdot \mathbf{1}^{\mathsf{T}})$.

To show $\mathrm{Rank}(\mathbf{I} - f \cdot \mathbf{1}^{\mathsf{T}}) \geq n - 1$, we can equivalently show that the nullspace of $(\mathbf{I} - f \cdot \mathbf{1}^{\mathsf{T}})$ has at most dimension one. suppose that $g$ is a nonzero vector in $\mathbb{R}^n$ that satisfies the equation:

$$(\mathbf{I} - f\mathbf{1}^{\mathsf{T}})g = 0 \iff g = f \cdot \mathbf{1}^{\mathsf{T}}g,$$

which also implies $\mathbf{1}^{\mathsf{T}} \cdot g = \mathbf{1}^{\mathsf{T}} \cdot f \cdot \mathbf{1}^{\mathsf{T}}g$, or $\mathbf{1}^{\mathsf{T}} \cdot f = 1$. Therefore, there are two cases to consider:

- If the value of $d$ satisfies $\mathbf{1}^{\mathsf{T}} \cdot f = \mathbf{1}^{\mathsf{T}}\mathbf{K}^{-1} \cdot d = 1$, the nullspace of $(\mathbf{I} - f\mathbf{1}^{\mathsf{T}})$ is span$(f)$, which has dimension one, and accordingly $\mathrm{Rank}(\mathbf{I} - f \cdot \mathbf{1}^{\mathsf{T}}) = \mathrm{Rank}(\mathring{\mathbf{K}}) = n - 1$.

35th Conference on Neural Information Processing Systems (NeurIPS 2021), Sydney, Australia.

- If the value of $d$ does not satisfy $\mathbf{1}^\intercal \mathbf{K}^{-1} \cdot d = 1$, the nullspace of $(\mathbf{I} - f\mathbf{1}^\intercal)$ has dimension zero, and consequently $\text{Rank}(\mathbf{I} - f \cdot \mathbf{1}^\intercal) = \text{Rank}(\overset{\circ}{\boldsymbol{K}}) = n$.

$\square$

We are now ready to present Theorem 1.

**Theorem 1** *Suppose we have observed* $\mathbf{X}$ *generated according to the mixing procedure (4) in a number of domains,* $t = 1, 2, ..., T$. *Assume that* $\varepsilon_i$ *are uncorrelated in each domain and that their variances in domain $t$, denoted by* $\sigma_{ti}^2$, *change independently across domains in the sense that* $\mathbf{S}$, *whose* $(i, t)$*th entry is* $\sigma_{ti}^2$, *has full column rank. Further assume that each $n$ columns of* $\mathbf{M}_\mathcal{X}$ *are linearly independent and that* $p \leq 2n - 2$. *Then if* $\mathbf{X}$ *admits a model*

$$\mathbf{X} = \tilde{\mathbf{M}}_\mathcal{X}\tilde{\varepsilon}, \tag{16}$$

*where* $\tilde{\varepsilon}$ *also follows the above assumption on* $\varepsilon$, *every column of* $\tilde{\mathbf{M}}_\mathcal{X}$ *must be proportional to a column of* $\mathbf{M}_\mathcal{X}$ *and vice versa.*

*Proof.* Let $\sigma_{ti}^2$ be the variance of $\tilde{\varepsilon}_i$ in the $t$th domain. Let $S_t$ be the diagonal matrix with $\sigma_{t1}^2, \sigma_{t2}^2, ..., \sigma_{tp}^2$ on its diagonal, and similarly for $\tilde{S}_t$. Let $\tilde{\mathbf{S}}$ be the matrix with $\tilde{\sigma}_{ti}^2$ as its $(i, t)$th entry. Denote by $\mathbf{M}_\mathcal{X}^i$ the $i$th column of $\mathbf{M}_\mathcal{X}$, and similarly for $\tilde{\mathbf{M}}_\mathcal{X}^i$. In the $t$-th domain the two mixing models imply the same distribution, or more specifically, the same covariance matrix, of $\mathbf{X}$. That is, in the $t$-th domain,

$$\text{Cov}(\mathbf{X}_t) = \mathbf{M}_\mathcal{X} S_t \mathbf{M}_\mathcal{X}^\intercal = \tilde{\mathbf{M}}_\mathcal{X} \tilde{S}_t \tilde{\mathbf{M}}_\mathcal{X}^\intercal, \quad \text{or equivalently,} \tag{17}$$

$$\text{Cov}(\mathbf{X}_t) = \sum_{i=1}^p \sigma_{ti}^2 \mathbf{M}_\mathcal{X}^i \mathbf{M}_\mathcal{X}^{i\intercal} = \sum_{i=1}^p \tilde{\sigma}_{ti}^2 \tilde{\mathbf{M}}_\mathcal{X}^i \tilde{\mathbf{M}}_\mathcal{X}^{i\intercal}. \tag{18}$$

It can also be written as

$$\sum_{i=1}^p \sigma_{ti}^2 \mathbf{M}_\mathcal{X}^i \otimes \mathbf{M}_\mathcal{X}^i = \sum_{i=1}^p \tilde{\sigma}_{ti}^2 \tilde{\mathbf{M}}_\mathcal{X}^i \otimes \tilde{\mathbf{M}}_\mathcal{X}^i, \text{ or in matrix form,}$$

$$(\mathbf{M}_\mathcal{X} \odot \mathbf{M}_\mathcal{X}) \cdot \mathbf{S} = (\tilde{\mathbf{M}}_\mathcal{X} \odot \tilde{\mathbf{M}}_\mathcal{X}) \cdot \tilde{\mathbf{S}}, \tag{19}$$

where $\otimes$ denotes the Kronecker product and $\odot$ the Khatri–Rao (column-wise Kronecker) product, i.e., $\tilde{\mathbf{M}}_\mathcal{X} \odot \tilde{\mathbf{M}}_\mathcal{X} = [\mathbf{M}_\mathcal{X}^1 \otimes \mathbf{M}_\mathcal{X}^1, \ \mathbf{M}_\mathcal{X}^2 \otimes \mathbf{M}_{\mathcal{X}2}, ..., \mathbf{M}_\mathcal{X}^p \otimes \mathbf{M}_\mathcal{X}^p]$.

Since $\mathbf{S}$ has full column rank, we can select $p$ columns from it, corresponding to $p$ domains, that form a full rank matrix. Let this matrix be $\mathbf{S}^*$. Similarly we have $\tilde{\mathbf{S}}^*$ corresponding to the alternative model (16), corresponding to the same $p$ domains. Equation (19) then implies

$$(\mathbf{M}_\mathcal{X} \odot \mathbf{M}_\mathcal{X}) \cdot \mathbf{S}^* = (\tilde{\mathbf{M}}_\mathcal{X} \odot \tilde{\mathbf{M}}_\mathcal{X}) \cdot \tilde{\mathbf{S}}^*, \tag{20}$$

We will use the concept Kruskal-rank [4], denoted by $\text{Rank}_k$; the Kruskal-rank of a matrix $K$ is the maximum number of $l$ such that every $l$ columns of $K$ are linearly independent. Bear in mind that each $n$ columns of $\mathbf{M}_\mathcal{X}$ are linear independent (i.e., $\text{Rank}_k(\mathbf{M}_\mathcal{X}) = n$) and that $p \leq 2n - 2$. Lemma 1 by Sidiropoulos et al. [5] then implies that the rank of $\mathbf{M}_\mathcal{X} \odot \mathbf{M}_\mathcal{X}$ is larger than or equal to $\min(2n - 1, p) = p$. That is, $\mathbf{M}_\mathcal{X} \odot \mathbf{M}_\mathcal{X}$ has full column rank. Further because $\mathbf{S}^*$ has full rank, (20) implies that $\tilde{\mathbf{S}}^*$ has full rank and that $\tilde{\mathbf{M}}_\mathcal{X} \odot \tilde{\mathbf{M}}_\mathcal{X}$ has full column rank.

Right-multiplying both sides of (20) by $\tilde{\mathbf{S}}^{*-1}$ and let $\mathbf{Q} := \mathbf{S}^* \cdot \tilde{\mathbf{S}}^{*-1}$, one will get

$$(\tilde{\mathbf{M}}_\mathcal{X} \odot \tilde{\mathbf{M}}_\mathcal{X}) = (\mathbf{M}_\mathcal{X} \odot \mathbf{M}_\mathcal{X}) \cdot \mathbf{Q}. \tag{21}$$

We shall then show that $\mathbf{Q}$ must be a generalized permutation matrix and hence the columns of $\tilde{\mathbf{M}}_\mathcal{X}$ are a permuted and scaled version of those of $\mathbf{M}_\mathcal{X}$.

Without loss of generality, let us consider the first column of the matrices on both sides of (21), and let $q_{i1}$ be the $(i, 1)$th entry of $\mathbf{Q}$. We have

$$\tilde{\mathbf{M}}_\mathcal{X}^1 \otimes \tilde{\mathbf{M}}_\mathcal{X}^1 = (\mathbf{M}_\mathcal{X} \odot \mathbf{M}_\mathcal{X}) \cdot \mathbf{Q}^1 = \sum_{i=1}^p q_{i1} \cdot (\mathbf{M}_\mathcal{X}^i \otimes \mathbf{M}_\mathcal{X}^i), \tag{22}$$

where $q_{i1}$ cannot be zero for all $i$. Since each $n$ columns of $\mathbf{M}_\mathcal{X}$ are linearly independent, it cannot contain a zero column. Suppose $\mathbf{M}^i_{\mathcal{X}k}$, the $(k,i)$th entry of $\mathbf{M}_\mathcal{X}$, is nonzero. According to the specific structure of the Kronecker product $\tilde{\mathbf{M}}^1_\mathcal{X} \otimes \tilde{\mathbf{M}}^1_\mathcal{X}$ in (22), we know that there must exist a non-zero vector $d \in \mathbb{R}^n$ such that the RHS satisfies

$$\sum_{i=1}^{p} q_{i1} \cdot (\mathbf{M}^i_\mathcal{X} \otimes \mathbf{M}^i_\mathcal{X}) = d \otimes \big( \sum_{i=1}^{p} q_{i1} \cdot (\mathbf{M}^i_{\mathcal{X}k} \cdot \mathbf{M}^i_\mathcal{X}) \big) = \sum_{i=1}^{p} q_{i1} \cdot \big((\mathbf{M}^i_{\mathcal{X}k} \cdot d) \otimes \mathbf{M}^i_\mathcal{X}\big)$$

$$\Longrightarrow \sum_{i=1}^{p} q_{i1} \cdot \big((\mathbf{M}^i_\mathcal{X} - \mathbf{M}^i_{\mathcal{X}k} \cdot d) \otimes \mathbf{M}^i_\mathcal{X}\big) = 0$$

$$\Longrightarrow (\mathring{\mathbf{M}}_\mathcal{X} \odot \mathbf{M}_\mathcal{X})\mathbf{Q}^1 = 0, \tag{23}$$

where $\mathring{\mathbf{M}}_\mathcal{X}$ is a $n \times n$ matrix with $(\mathbf{M}^i_\mathcal{X} - \mathbf{M}^i_{\mathcal{X}k} \cdot d)$ as its $i$-th column, i.e.,

$$\mathring{\mathbf{M}}_\mathcal{X} = \mathbf{M}_\mathcal{X} - \mathbf{M}^i_{\mathcal{X}k} \cdot d \cdot \mathbf{1}^T.$$

We are now about to show that in order for (23) to hold, $q_{i1} \neq 0$ for one and only one $i = 1, 2, ..., p$.

There are two cases to consider:

- Suppose one column of $\mathring{\mathbf{M}}_\mathcal{X}$ is zero. Note that since $\mathrm{Rank}_k(\mathbf{M}_\mathcal{X}) = n$, each pair of its columns are linearly independent, so there is only one zero column in $\mathring{\mathbf{M}}_\mathcal{X}$. Let the $r$-th column of $\mathring{\mathbf{M}}_\mathcal{X}$ be zero. Denote by $\mathring{\mathbf{M}}^{(-r)}_\mathcal{X}$ the matrix obtained by removing the $r$-th column from $\mathring{\mathbf{M}}_\mathcal{X}$, and similarly for $\mathbf{M}^{(-r)}_\mathcal{X}$. Let $\mathbf{Q}^1_{(-r)}$ be the vector obtained by removing the $r$-th entry from the vector $\mathbf{Q}^1$. According to Lemma 5, each $n$ columns of $\mathring{\mathbf{M}}^{(-r)}_\mathcal{X}$ have rank at least $n-1$, so $\mathrm{Rank}(\mathring{\mathbf{M}}^{(-r)}_\mathcal{X}) \leq n-1$. At the same time, $\mathrm{Rank}_k(\mathbf{M}^{(-r)}_\mathcal{X}) = n$. Moreover, $\mathring{\mathbf{M}}^{(-r)}_\mathcal{X} \odot \mathbf{M}^{(-r)}_\mathcal{X}$ has $p-1$ columns. Hence $\mathrm{Rank}(\mathring{\mathbf{M}}^{(-r)}_\mathcal{X}) + \mathrm{Rank}_k(\mathbf{M}^{(-r)}_\mathcal{X}) \geq n-1+n = 2n-1 \geq (p-1)+1$ because it is assumed that $p \leq 2n-2$. Lemma 1 by Guo et al. [6] then implies that $\mathring{\mathbf{M}}^{(-r)}_\mathcal{X} \odot \mathbf{M}^{(-r)}_\mathcal{X}$ has full column rank. On the other hand, (23) becomes

$$q_{r1} \cdot 0 + (\mathring{\mathbf{M}}^{(-r)}_\mathcal{X} \odot \mathbf{M}^{(-r)}_\mathcal{X})\mathbf{Q}^1_{(-r)} = (\mathring{\mathbf{M}}^{(-r)}_\mathcal{X} \odot \mathbf{M}^{(-r)}_\mathcal{X})\mathbf{Q}^1_{(-r)} = 0.$$

  Consequently, $\mathbf{Q}^1_{(-r)}$ is a zero vector because $\mathring{\mathbf{M}}^{(-r)}_\mathcal{X} \odot \mathbf{M}^{(-r)}_\mathcal{X}$ has full column rank. That is, only $q_{r1}$ is non-zero. Then (22) tells us that

$$\tilde{\mathbf{M}}^1_\mathcal{X} \otimes \tilde{\mathbf{M}}^1_\mathcal{X} = q_{r1} \cdot (\mathbf{M}^r_\mathcal{X} \otimes \mathbf{M}^r_\mathcal{X}).$$

  Hence, $\tilde{\mathbf{M}}^1_\mathcal{X}$ is a scaled version of $\mathbf{M}^r_\mathcal{X}$.

- Suppose no column of $\mathring{\mathbf{M}}_\mathcal{X}$ is zero. According to Lemma 5, each $n$ columns of $\mathring{\mathbf{M}}_\mathcal{X}$ have rank at least $n-1$, so $\mathrm{Rank}(\mathring{\mathbf{M}}_\mathcal{X}) \leq n-1$. Remember $\mathrm{Rank}_k(\mathbf{M}^{(-r)}_\mathcal{X}) = n$ and $\mathring{\mathbf{M}}_\mathcal{X} \odot \mathbf{M}_\mathcal{X}$ has $p$ columns. Hence $\mathrm{Rank}(\mathring{\mathbf{M}}_\mathcal{X}) + \mathrm{Rank}_k(\mathbf{M}_\mathcal{X}) \geq n-1+n = 2n-1 \geq p+1$ because $p \leq 2n-2$. Again, Lemma 1 by Guo et al. [6] indicates that $\mathring{\mathbf{M}}_\mathcal{X} \odot \mathbf{M}_\mathcal{X}$ has full column rank. Hence, in order for (23) to hold, $\mathbf{Q}^1$ must be a zero vector, leading to a contradiction.

Therefore, $\tilde{\mathbf{M}}^1_\mathcal{X}$ is a scaled version of $\mathbf{M}^r_\mathcal{X}$. Similarly $\tilde{\mathbf{M}}^2_\mathcal{X}$ is a scaled version of $\mathbf{M}^{r'}_\mathcal{X}$, and so on. Because $\tilde{\mathbf{M}}_\mathcal{X} \odot \tilde{\mathbf{M}}_\mathcal{X}$ has full column rank, different columns of $\tilde{\mathbf{M}}^1_\mathcal{X}$ must correspond to different columns of $\mathbf{M}_\mathcal{X}$. Further because of the symmetry between the two models (4) and (16), every column of $\tilde{\mathbf{M}}_\mathcal{X}$ must be proportional to a column of $\mathbf{M}_\mathcal{X}$ and vice versa.

$\square$

## 1.2 Proof of Theorem 2

**Theorem 2.** *If* $\mathbf{F}$ *is identified up to trivialities, then the graph induced by* $\mathbf{F}$ *satisfies the bottleneck and strong non-redundancy conditions.*

We prove this for the two conditions separately. Throughout, we use $\mathbf{F}$, $\mathbf{M}$, and $\mathrm{Ch}(\cdot)$ to refer to the relevant components of the true causal system; and $\hat{\mathbf{F}}$, $\hat{\mathbf{M}}$, and $\mathrm{Ch}'(\cdot)$ to refer to the relevant components of an alternative causal system which we will construct. $\mathbf{R}$ is as described in Proposition 3.

*Bottleneck condition:* Let $B \neq \mathrm{Ch}(V_i)$ be a minimal bottleneck from $\mathrm{Ch}(V_i)$ to $\mathcal{X}$. Define

$$\hat{\mathbf{F}}^i = \mathbf{I}_{\mathcal{V}}^B \mathbf{R}_B^i$$

so that $\hat{\mathbf{F}}_j^i = \left[\mathbf{R}_B^i\right]_j$ if $V_j \in B$ and $\hat{\mathbf{F}}_j^i = 0$ otherwise. Clearly, $(\mathbf{I} - \hat{\mathbf{F}})$ is invertible whenever $(\mathbf{I} - \mathbf{F})$ is. Moreover,

$$\mathbf{M}_{\mathcal{X}}\hat{\mathbf{F}}^i = \mathbf{M}_{\mathcal{X}}^B \mathbf{R}_B^{\mathrm{Ch}(V_i)} \mathbf{F}_{\mathrm{Ch}(V_i)}^i = \mathbf{M}_{\mathcal{X}}^{\mathrm{Ch}(V_i)} \mathbf{F}_{\mathrm{Ch}(V_i)}^i = \mathbf{M}_{\mathcal{X}} \mathbf{F}^i.$$

So, since $\mathbf{M}_{\mathcal{X}}\hat{\mathbf{F}} = \mathbf{M}_{\mathcal{X}}\mathbf{F} = (\mathbf{M} - \mathbf{I})_{\mathcal{X}}$, (3) shows that $(\mathbf{I} - \hat{\mathbf{F}})_{\mathcal{X}}^{-1} = \mathbf{M}_{\mathcal{X}}$. Thus $\hat{\mathbf{F}}$ generates $\mathcal{M}$. Furthermore, $\|\hat{\mathbf{F}}\|_0 \leq \|\mathbf{F}\|_0$ since $B$ is assumed to be minimal, so that $\hat{\mathbf{F}} \in \mathcal{F}$. Therefore, since $\mathbf{F}$ and $\hat{\mathbf{F}}$ induce different DAGs when $B \neq \mathrm{Ch}(V_i)$, $\mathbf{F}$ is not identified up to trivialities.

*Parental non-redundancy:* If $L_i \to V_j$, define $\mathbf{P}$ as the identity matrix with the $i$-th and $j$-th columns switched; $\mathbf{D}$ as the diagonal matrix with $D_i^i = 1/F_j^i$ and ones on the rest of the diagonal; and further

$$\hat{\mathbf{M}}_{\mathcal{X}} = \mathbf{M}_{\mathcal{X}}\mathbf{D}\mathbf{P},$$

$$\hat{\mathbf{F}}^i = \mathbf{I}^j + \mathbf{I}^i - \left(F_j^i\right)^{-1}\mathbf{P}\mathbf{F}^i,$$

$$\hat{\mathbf{F}}^j = \mathbf{P}\left[\left(F_j^i\right)^{-1}\mathbf{F}^i + (\mathbf{F} - \mathbf{I})^j\right],$$

$$\hat{\mathbf{F}}^k = \mathbf{P}\mathbf{D}^{-1}\mathbf{F}^k \text{ for all } k \notin \{i, j\},$$

so that $\mathrm{Ch}'(L_i) = \mathrm{Ch}(L_i)$ and $\mathrm{Ch}'(V_j) \subseteq \mathrm{Ch}(V_j)$, but with weights $\hat{\mathbf{F}}^i \not\propto \mathbf{F}^i$ and $\hat{\mathbf{F}}^j \not\propto \mathbf{F}^j$. Moreover, for every other $V_k$, if $L_i \in \mathrm{Ch}(V_k)$, then $V_j \in \mathrm{Ch}'(V_k)$ and vice versa. Clearly $\|\hat{\mathbf{F}}\|_0 \leq \|\mathbf{F}\|_0$ if $L_i$ is a parental redundancy of $V_j$. Moreover, the resulting graph is acyclic whenever the true graph is acyclic. We compute:

$$\begin{aligned}
\hat{\mathbf{M}}_{\mathcal{X}}\hat{\mathbf{F}}^i &= \hat{\mathbf{M}}_{\mathcal{X}}\left[\mathbf{I}^j + \mathbf{I}^i - \left(F_j^i\right)^{-1}\mathbf{P}\mathbf{F}^i\right] \\
&= \mathbf{M}_{\mathcal{X}}\left(F_j^i\right)^{-1}\left[\mathbf{I}^i + F_j^i\mathbf{I}^j - \mathbf{F}^i\right] \\
&= \left(F_j^i\right)^{-1}\left[\mathbf{M}_{\mathcal{X}}^i + F_j^i\mathbf{M}_{\mathcal{X}}^j - \mathbf{M}_{\mathcal{X}}^i + \mathbf{I}_{\mathcal{X}}^i\right] \\
&= \mathbf{M}_{\mathcal{X}}^j \\
&= \hat{\mathbf{M}}_{\mathcal{X}}^i,
\end{aligned}$$

where we have used the fact that $\mathbf{I}_{\mathcal{X}}^i = 0$ since $L_i \in \mathcal{L}$. We further calculate

$$\begin{aligned}
\hat{\mathbf{M}}_{\mathcal{X}}\hat{\mathbf{F}}^j &= \mathbf{M}\mathbf{D}\left[\left(F_j^i\right)^{-1}\mathbf{F}^i + (\mathbf{F} - \mathbf{I})^j\right] \\
&= \left(F_j^i\right)^{-1}\mathbf{M}^i - \mathbf{I}^j \\
&= \hat{\mathbf{M}}^j - \mathbf{I}^j,
\end{aligned}$$

and finally

$$\hat{\mathbf{M}}_{\mathcal{X}}\hat{\mathbf{F}}^k = \mathbf{M}_{\mathcal{X}}\mathbf{F}^k$$

for $k \notin \{i, j\}$. Hence $\hat{\mathbf{M}}_{\mathcal{X}}\hat{\mathbf{F}} = (\mathbf{M} - \mathbf{I})_{\mathcal{X}}$. Rearranging, we see that $(\mathbf{I} - \hat{\mathbf{F}})_{\mathcal{X}}^{-1} = \hat{\mathbf{M}}_{\mathcal{X}} \in \mathcal{M}$. Because we have already argued that $\|\hat{\mathbf{F}}\|_0 \leq \|\mathbf{F}\|_0$ if $L_i$ is a parental redundancy of $V_j$, it follows that $\hat{\mathbf{F}} \in \mathcal{F}$ if $L_i$ is a parental redundancy of $V_j$. Therefore, due to the changes in causal scale, $\mathbf{F}$ is not identifiable up to trivialities.

*Co-parental non-redundancy:* If $L_i \not\to V_j$ and $V_k \in \mathrm{Ch}(L_i) \cap \mathrm{Ch}(V_j)$, define

$$\hat{\mathbf{F}}^j = \mathbf{F}^j + (\mathbf{I} - \mathbf{F})^i f,$$

where $f := \frac{F_k^j}{F_k^i}$. Then $\mathrm{Ch}'(V_j) = \mathrm{Ch}(V_j) \cup \{L_i\} \cup \mathrm{Ch}(L_i) - \{V_k\}$. If $L_i$ is a co-parental redundancy of $V_j$, then the resulting system is no denser than the original. Moreover, the resulting system is acyclic. Finally, we calculate:

$$\mathbf{M}_\mathcal{X} \hat{\mathbf{F}}^j = (\mathbf{M} - \mathbf{I})_\mathcal{X}^j + f \mathbf{I}_\mathcal{X}^i$$
$$= (\mathbf{M} - \mathbf{I})_\mathcal{X}^j,$$

since $L_i \in \mathcal{L}$. With $\hat{\mathbf{F}}^k = \mathbf{F}^k$ for all remaining $k \neq i$, (3) shows that $(\hat{\mathbf{F}} - \mathbf{I})_\mathcal{X}^{-1} = \mathbf{M}_\mathcal{X}$, so that $\hat{\mathbf{F}} \in \mathcal{F}$. But since $\mathbf{F}$ induces a different DAG, $\mathbf{F}$ is not identified up to trivialities.

**Remark:** Notice that parental non-redundancy is not necessary to identify the full causal DAG; if $L_i$ is a parental redundancy of $V_j$, and neither $L_i$ nor $V_j$ has any parent, then both the true system and the alternative system constructed in the proof of Theorem 2 will have the same skeleton. However, the alternative system is emphatically not a mere re-indexing and re-scaling of latent variables. In particular, if $V_j \in \mathcal{X}$, then the net effect of $X_j$ on $\mathbf{X}$ will not be identified.

### 1.3 Characterizing bottleneck faithfulness

In this section we show that the set of adjacency matrices, corresponding to a fixed graph, that are not bottleneck faithful is a proper algebraic subset of all adjacency matrices for that graph. That is, the property of being bottleneck faithful is a generic property, both in the sense that it holds on an dense open set and that the exception set is of Lebesgue measure zero. But first, we prove Proposition 1 from the main paper.

*Proof.* (Prop. 1) Decompose $\mathbf{M}_K^J = \mathbf{M}_K^B \mathbf{R}_B^J$ using Proposition 3, then $\mathrm{Rank}\left(\mathbf{M}_K^J\right) \leq |B|$. □

To formalize that bottleneck faithfulness is a generic property, let $G$ denote a graph (a DAG) with $p$ nodes and $n$ edges. A $p \times p$ adjacency matrix $\mathbf{F}$ that induces $G$ has $n$ nonzero entries, and we let $\mathbb{F}_G \subseteq \mathbb{R}^n$ denote the set of adjacency matrices that induce $G$ – with $\mathbb{F}_G$ regarded as a subset of $\mathbb{R}^n$. An algebraic subset of $\mathbb{F}_G$ is a set

$$A = \{\mathbf{F} \in \mathbb{F}_G \mid \mathrm{pol}(\mathbf{F}) = 0\}$$

where $\mathrm{pol}$ is a polynomial. If $\mathrm{pol}$ is not the zero polynomial, $A$ is a proper algebraic subset, and it is well known that $A$ is then nowhere dense and of Lebesgue measure zero. We will construct a non-zero polynomial that evaluates to 0 if and only if the adjacency matrix is not bottleneck faithful. To this end, we first show that for any graph there exists an adjacency matrix that is bottleneck faithful.

**Proposition 4.** *For any graph $G$ there exists $\mathbf{F} \in \mathbb{F}_G$ such that $\mathbf{F}$ is bottleneck faithful.*

*Proof.* The proof is by induction on the number of edges, $n$. Clearly for $n = 0$ we have bottleneck faithfulness.

For the induction step, let $n \geq 1$ and suppose there exists a bottleneck faithful adjacency matrix for any graph with less than $n$ edges. Let $G'$ be a graph with $n$ edges, let $V_i$ be a root node with $\mathrm{Ch}'(V_i) \neq \varnothing$ the children of $V_i$ in $G'$. Let $G$ be the subgraph of $G'$ with all edges out of $V_i$ removed, and let $\mathbf{F} \in \mathbb{F}_G$ denote a bottleneck faithful adjacency matrix for $G$. By choosing $F_l^i$ for $l \in \mathrm{Ch}'(V_i)$ we can regard $\mathbf{F} \in \mathbb{F}_{G'}$, and the objective is to choose $F_l^i$ such that $\mathbf{F}$ becomes bottleneck faithful for $G'$. In what follows, $\mathbf{M}$ denotes the mixing matrix for $\mathbf{F}$ on $G$, that is, when $F_l^i = 0$ for $l \in \mathrm{Ch}'(V_i)$, and $\mathbf{M}'$ denotes the mixing matrix on $G'$ for any choice of $F_l^i$ for $l \in \mathrm{Ch}'(V_i)$.

Given $J, K \subseteq \mathcal{V}$ we denote by $L_K^J$ the set of coefficients $F_l^i$ for $l \in \mathrm{Ch}'(V_i)$ for which bottleneck faithfulness for $G'$ is **violated** by $J$ and $K$.

If $V_i \notin J$, then since $V_i$ is a root in $G'$, bottleneck faithfulness holds for $G'$ from $J$ to $K$ – no matter the coefficients $F_l^i$ for $l \in \mathrm{Ch}'(V_i)$. Thus $L_K^J = \varnothing$.

If $V_i \in J$, we have that $\mathbf{M}_K^i = \mathbf{I}_K^i$, and

$$\mathbf{M}_K'^i = \mathbf{I}_K^i + \sum_{l \in \mathrm{Ch}'(V_i)} F_l^i \mathbf{M}_K^l \quad \text{and} \quad \mathbf{M}_K'^j = \mathbf{M}_K^j \text{ for } j \neq i.$$

There are two cases to consider.

1. *There is a minimal bottleneck, $B$, from $J$ to $K$ in $G$ such that all paths from $i$ to $K$ in $G'$ pass through $B$.*

   In this case, $B$ is also a minimal bottleneck in $G'$. Since $V_i$ is a root node, $\mathbf{M}_i^j = 0$ for all $j$, and replacing zero entries in the column $\mathbf{I}_K^i$ by possibly non-zero entries in the column $\mathbf{M}_K'^i$ cannot reduce the rank of $\mathbf{M}_K^J$. Thus no choice of $F_l^i$ for $l \in \mathrm{Ch}'(V_i)$ makes $J$ and $K$ violate bottleneck faithfulness for $G'$ and $L_K^J = \varnothing$.

2. *For any minimal bottleneck, $B$, from $J$ to $K$ in $G$ there is a path from $i$ to $K$ in $G'$ that does not pass through $B$.*

   Note that in this case, $V_i \notin K$ and $\mathbf{I}_K^i = 0$. Choose any minimal bottleneck, $B$, in $G$. Then $B \cup \{i\}$ is a minimal bottleneck in $G'$, and with $\mathrm{col}(\mathbf{M}_K^J)$ the column space of $\mathbf{M}_K^J$,

   $$L_K^J = \left\{ (F_l^i)_{l \in \mathrm{Ch}'(V_i)} \;\middle|\; \sum_{l \in \mathrm{Ch}'(V_i)} F_l^i \mathbf{M}_K^l \in \mathrm{col}(\mathbf{M}_K^J) \right\} \subseteq \mathbb{R}^{\mathrm{Ch}'(V_i)}.$$

   Note that $L_K^J$ is a linear subspace of $\mathbb{R}^{\mathrm{Ch}'(V_i)}$. Due to bottleneck faithfulness for $G$, $\mathrm{Rank}(M_K^{J \cup \mathrm{Ch}'(V_i)}) \geq |B| + 1$, or there would be a bottleneck from $J$ to $K$ in $G'$ of size $|B|$. This shows that $L_K^J$ is a **true** subspace.

The set $\bigcap_{J,K}(L_K^J)^c$ contains all valid choices of coefficients $F_l^i$ for $l \in \mathrm{Ch}'(V_i)$ such that $\mathbf{F}$ is bottleneck faithful for $G'$. Since all $L_K^J$ are true subspaces, their complements are open and dense and so is their intersection. It is, in particular, non-empty and contains an element with $F_l^i \neq 0$ for all $l \in \mathrm{Ch}'(V_i)$. $\qquad\square$

**Proposition 5.** *Let $G$ be a graph with $n$ edges. The set*

$$A = \{\mathbf{F} \in \mathbb{F}_G \mid \mathbf{F} \text{ is bottleneck faithful for } G\}$$

*is a proper algebraic subset of $\mathbb{R}^n$. In particular, a generic adjacency matrix $\mathbf{F}$ is bottleneck faithful.*

*Proof.* Recall that the mixing matrix for $\mathbf{F}$ is

$$\mathbf{M} = (I - \mathbf{F})^{-1} = I + \mathbf{F} + \ldots + \mathbf{F}^p,$$

thus the entries in $\mathbf{M}$ are polynomials in the coefficients in $\mathbf{F}$. For any $J$ and $K$, let $b_K^J$ denote the size of a minimal bottleneck from $J$ to $K$, let

$$\mathbb{H}_K^J = \{\mathbf{H} \mid \mathbf{H} \text{ is a } b_K^J \times b_K^J \text{ submatrix of } \mathbf{M}_K^J\},$$

and define

$$\mathrm{pol}_K^J(\mathbf{F}) = \sum_{\mathbf{H} \in \mathbb{H}_K^J} \det(\mathbf{H})^2.$$

Clearly, $\mathrm{pol}_K^J$ is a polynomial in the coefficients of $\mathbf{F}$, $\mathrm{pol}_K^J(\mathbf{F}) = 0$ if and only if bottleneck faithfulness is violated for $\mathbf{F}$ by $J$ and $K$, and

$$A = \left\{\mathbf{F} \in \mathbb{F}_G \;\middle|\; \prod_{J \subseteq \mathcal{V}, K \subseteq \mathcal{V}} \mathrm{pol}_K^J(\mathbf{F}) = 0 \right\}$$

is the set of adjacency matrices that are not bottleneck faithful. By Proposition 4 it is non-empty, thus the defining polynomial is not the zero polynomial and $A$ is a proper algebraic subset. $\qquad\square$

### 1.4 Proofs of Theorem 3 and its associated lemmas

First we prove a useful result not in the main text.

**Proposition 6.** *Suppose a partially observed DAG satisfies the bottleneck condition and generalized non-redundancy. For every $V_i, V_j$, $\mathrm{Ch}(V_i)$ is a bottleneck from $\mathrm{Ch}(V_j)$ to $\mathcal{X}$ if and only if $V_i = V_j$.*

*Proof.* The backward direction is obvious. Conversely, define

$$S := \{V_k \in \text{Desc}(V_j) - \text{Ch}(V_i) : \text{Ch}(V_i) \text{ is a bottleneck from } V_k \text{ to } \mathcal{X}\}.$$

Obviously $S \cap \mathcal{X} = \varnothing$, and if $S = \varnothing$, then $\text{Ch}(V_i)$ is not a bottleneck from $V_j$ to $X$. Hence, by acyclicity, there is an $L_n \in S$ which has no descendent in $S$. Therefore $\text{Ch}(L_n) \subseteq \text{Ch}(V_i) \cup \{V_i\}$. $\qquad\square$

As indicated in the main paper, we assume in Lemmas 1-4 and Theorem 3 that the bottleneck condition, strong non-redundancy, and bottleneck faithfulness hold, and that $\mathcal{M}$ is identifiable.

### 1.4.1 Proof of Lemma 1

**Lemma 1.** *Let $J = \text{Ch}(V_i)$. Then the unique solution to (8) is given by $\mathbf{x} = \mathbf{F}^i_{\text{Ch}(V_i)}$.*

*Proof.* For uniqueness, notice that

$$\text{Rank}\left(\mathbf{M}^{\text{Ch}(V_i)}_{\mathcal{X}}\right) \geq \text{Rank}\left(\mathbf{M}^{\text{Ch}(V_i)}_{\mathcal{X}-\{V_i\}}\right) = |\text{Ch}(V_i)|,$$

with the equality following from the bottleneck condition and bottleneck faithfulness.

To see that $\mathbf{F}^i_J$ is a solution, notice that $\mathbf{M}^J_{\mathcal{X}}\mathbf{F}^i_J = \mathbf{M}_{\mathcal{X}}\mathbf{F}^i$, and use (3). $\qquad\square$

### 1.4.2 Proof of Lemma 2

**Lemma 2.** *Suppose $J \subseteq \mathcal{V} - \text{Anc}(V_i)$. If $(\mathbf{M} - \mathbf{I})^i_{\mathcal{X}} \in \text{Range}\left(\mathbf{M}^J_{\mathcal{X}}\right)$, then $|J| \geq |\text{Ch}(V_i)|$, with equality if and only if $J = \text{Ch}(V_i)$.*

*Proof.* Let $J \subseteq \mathcal{V} - \text{Anc}(V_i)$. The $i$-th row of (8) is satisfied trivially: $\mathbf{M}^J_i = 0$ when $J \subseteq \mathcal{V} - \text{Anc}(V_i)$, and $(\mathbf{M} - \mathbf{I})^i_i = 0$ by acyclicity. Thus (8) has a solution if and only if

$$\mathbf{M}^i_{\mathcal{X}-\{V_i\}} = \mathbf{M}^J_{\mathcal{X}-\{V_i\}}\mathbf{x}$$

has a solution. This can be factorized as

$$\mathbf{M}^B_{\mathcal{X}-\{V_i\}}\mathbf{R}^i_B = \mathbf{M}^B_{\mathcal{X}-\{V_i\}}\mathbf{R}^J_B\mathbf{x},$$

where $B$ is any minimal bottleneck from $\{V_i\} \cup J$ to $\mathcal{X} - \{V_i\}$. By bottleneck faithfulness,

$$\text{Rank}\left(\mathbf{M}^B_{\mathcal{X}-\{V_i\}}\right) = |B|$$

so that there is a solution to (8) if and only if

$$\mathbf{R}^i_B = \mathbf{R}^J_B\mathbf{x}.$$

We distinguish two cases: either $V_i \in B$, or $V_i \notin B$.

In the first case, $\mathbf{R}^i_B$ is a $B$-dimensional basis vector with $1$ in the $i$-th slot and $0$ elsewhere. Thus if there is a solution, $\mathbf{R}^J_i \neq 0$, so that $J$ has a path to $V_i$. Hence $J \cap \text{Anc}(V_i) \neq \varnothing$.

In the second case, bottleneck faithfulness and the fact that $B$ is a minimal bottleneck indicate that $|J| \geq |B|$. Noting that $B$ is a bottleneck from $\text{Ch}(V_i)$ to $\mathcal{X}$, we apply the bottleneck condition: $|B| \geq |\text{Ch}(V_i)|$ with equality if and only if $B = \text{Ch}(V_i)$. Moreover, for each $V_j \in J$, $\text{Ch}(V_i)$ is a bottleneck from $V_j$ to $\mathcal{X} - \{V_i\}$ if and only if $V_j \in \text{Ch}(V_i)$ by Proposition 6. Combining, $|J| \geq |B| \geq |\text{Ch}(V_i)|$ with equalities if and only if $J = B = \text{Ch}(V_i)$. $\qquad\square$

### 1.4.3 Proof of Lemma 3

**Lemma 3.** *For every $k \geq 0$, let $\mathcal{V}_k$ and $\mathcal{J}_k(V_i)$ be defined as in the main paper. Then $V_i \in \mathcal{V}_{k+1} - \mathcal{V}_k$ if and only if all of the following hold:*

    *1. $V_i \notin \mathcal{V}_k$,*

    *2. $|\text{Support}(\mathbf{M}^i) - \mathcal{X}_k| \leq 1$,*

3. $|\mathcal{J}_{k+1}(V_i)| = 1$, and

4. for all $V_j \neq V_i$ satisfying points 1 and 2, $\mathbf{M}^j_{\mathcal{X}_k} \notin \mathrm{Range}\left(\mathbf{M}^{J_k(V_i)}_{\mathcal{X}_k}\right)$.

*Proof.* Suppose $V_i \in \mathcal{V}_{k+1} - \mathcal{V}_k$. The first conjunct is obvious, the second conjunct follows by acyclicity, and the third conjunct follows from Lemma 2. For the fourth conjunct, take any other $V_j$, and let $B$ be a minimal bottleneck from $\{V_j\} \cup \mathrm{Ch}(V_i)$ to $\mathcal{X}_k$. Then $B$ is further a bottleneck from $\{V_j\} \cup \mathrm{Ch}(V_i)$ to $\mathcal{X}$. By Proposition 6, $B \neq \mathrm{Ch}(V_i)$, since $B$ is in particular a bottleneck from $V_j$ to $\mathcal{X}$. Hence, by the bottleneck condition, $|B| > |\mathrm{Ch}(V_i)|$, since $B$ is in particular a bottleneck from $\mathrm{Ch}(V_i)$ to $X$. Therefore,

$$\mathbf{R}^j_B = \mathbf{R}^{\mathrm{Ch}(L_i)}_B \mathbf{x}$$

has no solution due to bottleneck faithfulness, so that

$$\mathbf{M}^j_{\mathcal{X}_k} = \mathbf{M}^{\mathrm{Ch}(V_i)}_{\mathcal{X}_k} \mathbf{x}$$

also has no solution by bottleneck faithfulness applied to $\mathbf{M}^B_{\mathcal{X}_k}$.

Conversely, suppose $V_i \notin \mathcal{V}_{k+1} - \mathcal{V}_k$, and let $J \in \mathcal{J}_{k+1}(V_i)$. Clearly there exists some $V_j \in (\mathcal{V}_{k+1} - \mathcal{V}_k) \cap \mathrm{Desc}(V_i)$. Now, for any minimal bottleneck $B$ from $\{V_i\} \cup J$ to $\mathcal{X}_k$, $\mathbf{R}^i_B \in \mathrm{Range}(\mathbf{R}^J_B)$ by bottleneck faithfulness on $\mathbf{M}^B_{\mathcal{X}_k}$. Since $J \subseteq \mathcal{V}_k$, it follows that $B \subseteq \mathcal{V}_k$; otherwise some row of $\mathbf{R}^J_B$ is zero, proving that either $B$ is not minimal, or that $J$ does not admit a solution to (8). So in particular, $\mathrm{Anc}(V_j) \cap B = \varnothing$. Therefore, since $V_i$ has a path to $V_j$, and since $B$ is a bottleneck from $V_i$ to $\mathcal{X}_k$, $B$ is a bottleneck from $V_j$ to $\mathcal{X}_k$. Hence

$$\mathbf{M}^j_{\mathcal{X}_k} \in \mathrm{Range}\left(\mathbf{M}^B_{\mathcal{X}_k}\right) \subseteq \mathrm{Range}\left(\mathbf{M}^J_{\mathcal{X}_k}\right),$$

since $B$ is by definition a bottleneck from $J$ to $\mathcal{X}_k$. Because $V_j$ clearly satisfies conjuncts 1 and 2, the fourth conjunct is violated. $\square$

### 1.4.4 Proof of Lemma 4

**Lemma 4.** $X_i \in \mathcal{X}_{k+1}$ if and only if $X_i \in \mathcal{V}_{k+1}$ and $\mathrm{Support}\left(\mathbf{M}^i\right) - \mathcal{X}_k = \{i\}$.

*Proof.* This follows from definitions and acyclicity. $\square$

### 1.4.5 Proof of Theorem 3

**Theorem 3.** *Suppose $\mathbf{F}$ satisfies strong non-redundancy, bottleneck faithfulness, and the bottleneck condition. Then $\mathbf{F}$ is identifiable up to trivialities.*

*Proof.* Lemma 3 shows that $\mathbf{M}^{\mathcal{V}_k}_{\mathcal{X}}$ is identifiable from $\mathcal{M}$ up to permutation and scaling of columns, since neither it nor Lemma 2 upon which it relies makes any assumptions about the scaling or permutation of $\mathbf{M}$.

From here, Lemma 4 shows that $\mathbf{M}^X_{\mathcal{X}}$ is identifiable up to scaling for each $X \in \mathcal{X}$. But then $\mathbf{M}^X_{\mathcal{X}}$ is identifiable exactly since $\mathbf{M}^X_X = 1$ by acyclicity. Hence $\mathbf{M}^{\mathcal{X}}_{\mathcal{X}}$ is identifiable exactly, and $\mathbf{M}^{\mathcal{L}}_{\mathcal{X}}$ up to permutation and scaling of columns. In other words,

$$\tilde{\mathcal{M}} := \{\mathbf{M}_{\mathcal{X}}\mathbf{DP} : \mathbf{DP} \in \mathcal{DP}_p \text{ with } (\mathbf{DP})^{\mathcal{X}}_{\mathcal{X}} = \mathbf{I}\} \subset \mathcal{M}$$

is identifiable.

Fix any $\tilde{\mathbf{M}} \in \tilde{\mathcal{M}}$, and let $\mathbf{PD}$ satisfy $\tilde{\mathbf{M}}\mathbf{PD} = \mathbf{M}_{\mathcal{X}}$. Without loss of generality, reindex the latent variables so that $\mathbf{P} = \mathbf{I}$. Because $\mathcal{V}_k$ is identified for every $k$, apply Lemma 2 to conclude that $\tilde{\mathbf{M}}^{\mathrm{Ch}(V_i)}$ is identifiable for every $i$. Moreover, notice that $\tilde{\mathbf{M}}^i = \tilde{\mathbf{M}}^{\mathrm{Ch}(V_i)}\mathbf{x}$ if and only if $\mathbf{M}^i_{\mathcal{X}} = \mathbf{M}^{\mathrm{Ch}(V_i)}_{\mathcal{X}}\left[\mathbf{D}^{\mathrm{Ch}(V_i)}_{\mathrm{Ch}(V_i)}\mathbf{x}/D^i_i\right]$. By Lemma 1, this holds if and only if $\mathbf{x} = D^i_i(\mathbf{D}^{-1})^{\mathrm{Ch}(V_i)}_{\mathrm{Ch}(V_i)}\mathbf{F}^i_{\mathrm{Ch}(V_i)}$. Repeating for every $V_i$, $\tilde{\mathbf{F}} := \mathbf{D}^{-1}\mathbf{F}\mathbf{D}$ is identified from $\tilde{\mathbf{M}}$. $\square$

## 2 Detailed experimental results

### 2.1 BIC penalty

As indicated in the main paper, we created 10 causal systems for each of the graphs in Figure 5. The causal weights were drawn uniformly from $(-0.9, -0.5) \cup (0.5, 0.9)$, and the independent variances $\sigma^2_{t,i}$ were drawn independently and uniformly from $(0.5, 2.0)$. For each of the $T$ domains, $n$ samples were then simulated according to (12).

To estimate the full system, we enumerate all partially observed DAGs with three observed and at most two latent variables, and optimize all causal weights and noise variances using L-BFGS. As initialization, every covariance term and non-zero causal weight was set to 1. We then selected the graph with the lowest BIC:

$$\text{BIC}(\mathbf{F}, \Sigma) = -2\ell\ell(\mathbf{F}, \Sigma) + \|\mathbf{F}\|_0 \log(nT).$$

The table below summarizes the rate at which the correct skeleton was learned for each graph and each choice of $T \times n$.

| Graph | $1 \times 5000$ | $3 \times 1666$ | $5 \times 500$ | $5 \times 1000$ | $5 \times 10000$ |
|---|---|---|---|---|---|
| (i) | 10 | 10 | 10 | 10 | 10 |
| (ii) | 4 | 9 | 9 | 10 | 10 |
| (iii) | 2 | 10 | 9 | 8 | 9 |
| (iv) | 2 | 10 | 10 | 9 | 10 |
| (v) | 10 | 10 | 9 | 9 | 10 |
| (vi) | 1 | 7 | 4 | 6 | 8 |
| (vii) | 0 | 7 | 8 | 10 | 10 |
| (viii) | 0 | 7 | 8 | 9 | 10 |
| (ix) | 0 | 9 | 8 | 9 | 10 |
| (x) | 0 | 6 | 6 | 10 | 10 |
| (xi) | 0 | 9 | 6 | 9 | 10 |
| (xii) | 0 | 9 | 9 | 9 | 10 |
| (xiii) | 0 | 6 | 3 | 7 | 8 |
| (xiv) | 0 | 2 | 2 | 4 | 9 |
| (xv) | 0 | 0 | 1 | 3 | 6 |
| (xvi) | 2 | 9 | 8 | 9 | 10 |
| (xvii) | 0 | 0 | 1 | 3 | 5 |
| (xviii) | 0 | 5 | 7 | 8 | 10 |
| (xix) | 0 | 3 | 0 | 4 | 10 |
| (xx) | 0 | 5 | 2 | 6 | 10 |
| (xxi) | 0 | 3 | 1 | 2 | 8 |
| (xxii) | 0 | 5 | 0 | 5 | 8 |

Notice that the three sample sizes $1 \times 5000$, $3 \times 1666$, and $5 \times 1000$ all have the same total number of samples; any difference in performance is therefore attributable to the diversity of domains, and not to mere sample size.

Obviously these are not the only identifiable graphs; however, they are the only identifiable graphs up to re-indexing of variables. For example, graph (vii) has three versions: $X_1 \leftarrow L \rightarrow X_2$, $X_1 \leftarrow L \rightarrow X_3$, and $X_2 \leftarrow L \rightarrow X_3$. However, it is clearly sufficient to study the empirical recovery rate of only one of the three structures. Nevertheless, the exhaustive search was performed over all 1759 possible graphs with at most two latent variables; that is to say, for example, that the BIC optimization for graph (vii) included all three of these possibilities.

Recovery for graphs with exactly one latent and over 3 edges—graphs (xiii), (xiv), and (xv)—was relatively poor. In many incorrectly recovered graphs, the model of best fitting had an additional latent variable. In some sense, this gives the model an extra degree of freedom to approximate the covariances, by providing a larger overcomplete basis. However, only the number of edges was penalized in the L0 penalty, and not the number of latents. This was not possible in the case of (xii), as every graph with two latents has at least 4 edges, and so the BIC penalty was effective to prevent this. It is possible that this could be avoided by choosing the number of latent variables by a separate prior method, or by penalizing the number of latent variables. However, since this is not relevant

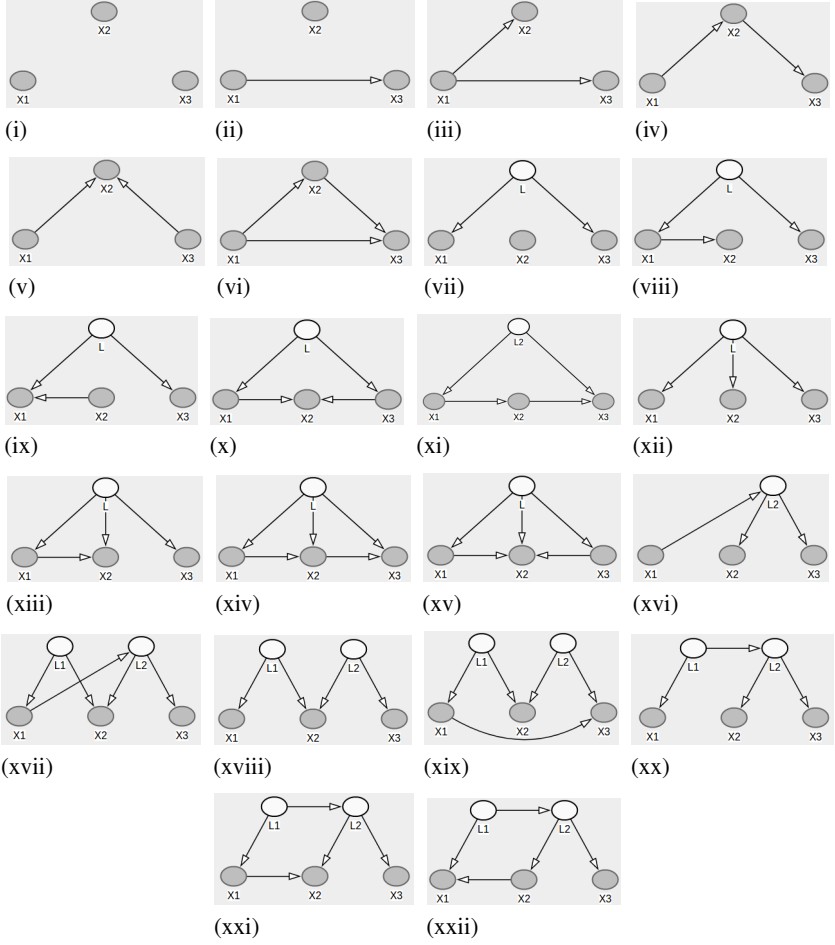

Figure 5: Identifiable graph skeletons, up to re-indexing of variables.

to enough graphs which are computationally admissible in an exhaustive search, we are not able to effectively study these cases.

The other graph with poor performance was (xvii). Note that in this case, with two latent variables and three measured variables, the mixing matrix $M_\mathcal{X}$ is not guaranteed to be recoverable by Theorem 1. (Recall that Theorem 1 gives sufficient identifibility conditions, which might not be necessary, and in the two cases the condition $p \le 2n - 2$ does not hold.) However, in every alternative graph, the recovered system had a very different mixing matrix—even in terms of sparsity patterns up to permutation of columns! We therefore attribute this indeterminacy to a non-identifiable mixing matrix in the heterogeneous case, and not to an unidentifiable graph structure in general (for example, in the single-domain non-Gaussian setting).

For the sake of reproducibility, we have included the code for this main experiment, along with instructions for how to generate these results.

## 2.2 Detailed experimental results: Unidentifiable graphs

Here we show detailed estimation results for the three equivalence classes of Figure 6. For each of graphs (i) through (ix) of Figure 6, we generated 15 adjacency matrices as in the identifiable experiments—that is, with weights drawn uniformly from $(-0.9, -0.5) \cup (0.5, 0.9)$. Further, we drew 1000 samples from 5 domains, with the variance of each noise term $\sigma_{t,i}^2$ drawn uniformly from the interval $(0.5, 2.0)$. We then optimized the log likelihood for each of the three systems in each equivalence class, and selected the model with the best optimized log likelihood. Since we are only showing that the graphs in each class are equivalent, we do not need to search over all possible latent

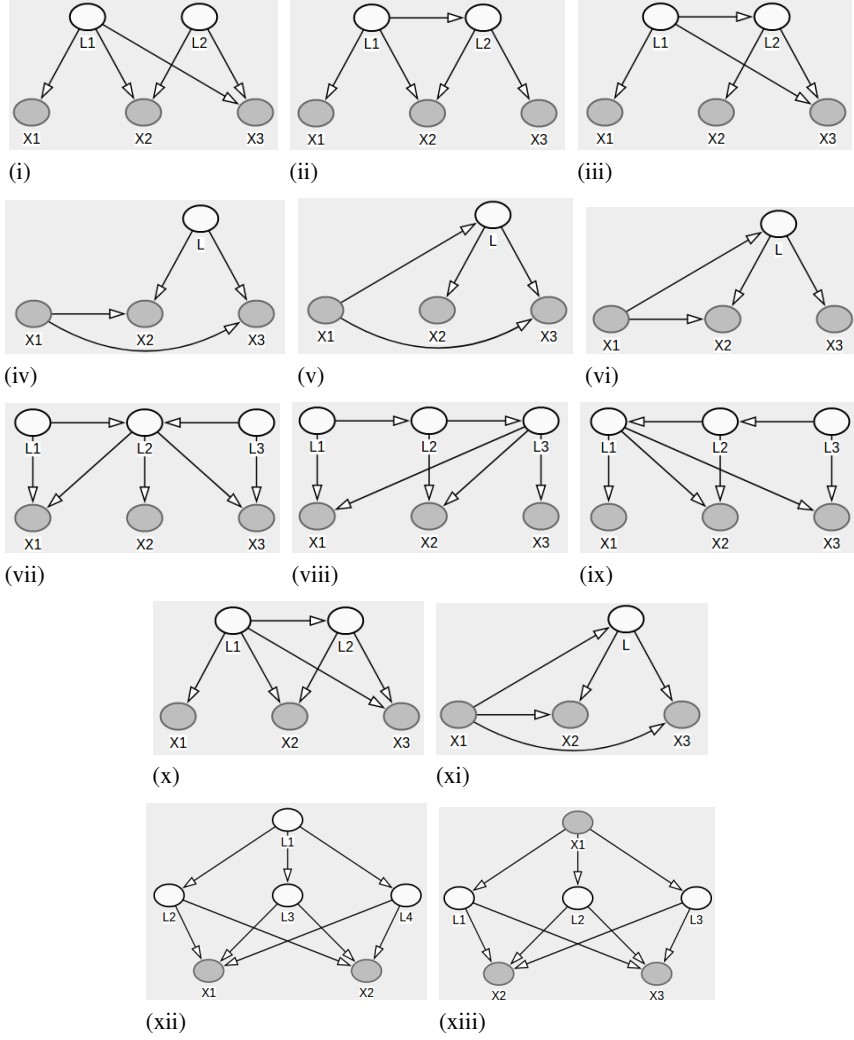

Figure 6: Three equivalence classes of graphs: {(i), (ii), (iii)} are equally sparse and observationally indistinguishable, as are {(iv), (v), (vi)} and {(vii), (viii), (ix)}. Moreover, (x) (xi) (xii) and (xiii) are not minimal, with (x) being observationally equivalent to (i) and (xi) being observationally equivalent to (iv). (xii) and (xiii) are discussed in Figure 1 of the main paper.

DAGs, but only over DAGs in the equivalence class. The charts below show the number of times each graph was chosen.

| True graph | Times (i) was selected | Times (ii) was selected | Times (iii) was selected |
|---|---|---|---|
| (i) | 6 | 3 | 6 |
| (ii) | 7 | 4 | 4 |
| (iii) | 7 | 1 | 7 |

| True graph | Times (iv) was selected | Times (v) was selected | Times (vi) was selected |
|---|---|---|---|
| (iv) | 8 | 3 | 4 |
| (v) | 6 | 4 | 5 |
| (vi) | 7 | 2 | 6 |

| True graph | Times (vii) was selected | Times (viii) was selected | Times (ix) was selected |
|---|---|---|---|
| (vii) | 2 | 5 | 8 |
| (viii) | 1 | 8 | 6 |
| (ix) | 1 | 7 | 7 |

In general, the difference in average log likelihood was on the same order as the convergence tolerance, $\|\nabla 2\ell\ell/n\|_\infty < 10^{-8}$, where $n$ is the total number of samples.

We ran a similar experiment for the non-minimal graphs (x)-(xiii). For each, 10 heterogeneous causal systems were generated, and 1000 samples were simulated from each of the $T = 5$ domains. For every overly dense graph, on all 10/10 trials, a sparser and observationally equivalent graph received a lower BIC score than the true overly dense one.

## 2.3 Detailed experimental results: L1 penalty

In this section, we show exact results for a random system generated with skeleton (xviii). Results for other partially observed graphs are similar.

The true causal model is given by:

$$
\begin{pmatrix} L_1 \\ L_2 \\ X_1 \\ X_2 \\ X_3 \end{pmatrix} =
\begin{bmatrix}
0 & 0 & 0 & 0 & 0 \\
0 & 0 & 0 & 0 & 0 \\
0.82 & 0 & 0 & 0 & 0 \\
0.53 & 0.51 & 0 & 0 & 0 \\
0 & 0.82 & 0 & 0 & 0
\end{bmatrix}
\begin{pmatrix} L_1 \\ L_2 \\ X_1 \\ X_2 \\ X_3 \end{pmatrix} + \varepsilon.
$$

As in the tests of the BIC algorithm, we used 5 domains. In the $t$-th domain, $\varepsilon \sim \mathcal{N}(0, \Sigma_t)$ for diagonal $\Sigma_t$. The variances for the $t$-th domain (i.e. the diagonal entries of $\Sigma_t$) are listed in the $t$-th row of the matrix below:

$$
\mathbf{S} =
\begin{bmatrix}
1.45 & 0.71 & 1.91 & 1.28 & 1.12 \\
0.89 & 1.66 & 1.18 & 1.35 & 0.52 \\
1.42 & 1.41 & 1.42 & 1.91 & 1.52 \\
1.03 & 1.15 & 1.54 & 0.59 & 1.50 \\
1.50 & 0.81 & 0.69 & 0.97 & 1.04
\end{bmatrix}.
$$

All coefficients were randomly drawn by the same method used for the BIC-simulation studies.

We simulated 1000 observations for each of the 5 domains, and then estimated the adjacency matrix by minimizing

$$
-2\ell\ell(\mathbf{F}, \Sigma)/(nT) + \lambda \sum |F_{i,j}|
$$

as in (15), subject to $\sigma_{t,i} \in (0.1, 2.0)$ for each $t \in \{1, ..., 5\}$ and $i \in \mathcal{L}$. It is necessary to bound each of the latent $\sigma$, because otherwise it would be possible to evade the L1 penalty by making $\mathbf{F}^{\mathcal{L}}$ very small but $\Sigma^{\mathcal{L}}$ very large. However, to give the L1 optimizer the fairest chance of finding the true system, we constrained the latent values of $\sigma$ with the same upper bound as $\Sigma$ was generated with.

Because this is a non-convex objective, we ran L-BFGS-B from 10 random initializations, and used the point which best optimized (15).

Below we show the best-fitting adjacency matrix for various choices of $\lambda$. Recovered edges with strength in $(-0.1, 0.1)$ were pruned.

$\lambda = .5$; our procedure returned:

$$
\begin{bmatrix}
0 & 0 & 0 & 0 & 0 \\
0 & 0 & 0 & 0 & 0 \\
0 & 0 & 0 & 0 & 0 \\
0 & 0 & 0 & 0 & 0 \\
0 & 0 & 0 & 0 & 0
\end{bmatrix}
$$

$\lambda = .1$; our procedure returned:

$$
\begin{bmatrix}
0 & 0 & 0 & 0 & 0 \\
0 & 0 & 0 & 0 & 0 \\
0 & 0 & 0 & 0 & 0 \\
0 & 0 & 0.20 & 0 & 0 \\
0 & 0 & 0 & 0.23 & 0
\end{bmatrix}
$$

$\lambda = .015$; our procedure returned:

$$\begin{bmatrix} 0 & 0 & 0 & 0 & 0 \\ 0 & 0 & 0 & 0 & 0 \\ 0 & 0 & 0 & 0 & 0 \\ 0 & 0.37 & 0.24 & 0 & 0 \\ 0 & 0.31 & 0 & 0.19 & 0 \end{bmatrix}$$

Interestingly, at this choice of $\lambda$, the true graph was no longer a local minimum; with the truth as initialization, the optimizer moves to

$$\begin{bmatrix} 0 & 0 & 0 & 0 & 0 \\ 0 & 0 & 0 & 0 & 0 \\ 0.45 & 0 & 0 & 0 & 0 \\ 0.39 & 0.36 & 0.12 & 0 & 0 \\ 0 & 0.41 & 0 & 0.16 & 0 \end{bmatrix}$$

which has a much smaller L1 penalty than the true system.

$\lambda = .01$; our procedure returned:

$$\begin{bmatrix} 0 & 0 & 0 & 0 & 0 \\ 0 & 0 & 0 & 0 & 0 \\ 0.39 & 0 & 0 & 0 & 0 \\ 0.38 & 0.37 & 0.16 & 0 & 0 \\ 0 & 0.32 & 0 & 0.21 & 0 \end{bmatrix}$$

With the truth as initialization, the optimizer moves to an adjacency matrix with similar support. Again, these systems incur a smaller L1 penalty than the true system, even though they are denser than the true system.

Similar results for each choice of $\lambda$ are obtained with 10 000 samples in each domain.