# OpenReview forum: "Identification of Partially Observed Linear Causal Models: Graphical Conditions for the Non-Gaussian and Heterogeneous Cases"
_NeurIPS.cc/2021/Conference — NeurIPS 2021 Poster_

### Official Review · Reviewer_FkNC · 2021-07-07

**Rating:** 5
**Confidence:** 2

**Summary:**

This paper considers one of important problems that under what condition a directed acyclic graphical model, especially linear structural equatioon models, can be inferred without the causal sufficientcy assumption. Hence, the proposed method provide a novel approch that can cope with learning linear structural equation models even when there are several latent variables. The key conditions are (i) sparsest true model, (ii) minimal bottleneck, (iii) strong non-redundancy, (iv) bottleneck faithfulness. It tried to explain a lot of new concepts with examples, and to compare the proposed idea to the existing conditions such as faithfulness.

**Ethical Concerns:**

None.

**Limitations And Societal Impact:**

Not related.

**Main Review:**

I enjoyed the idea and I think the main problem of learning latent causal graphical models is very important; however, this paper should clarify some necessary definitions, concepts and examples; otherwise I cannot properly evaluate its originality and contribution.

line 41/43/84: It is unclear that
"\epsilon is a vector of independent noise variables", "\epsilon_i are uncorrelated within each domain, ...", and "\epsilon_i are not necessarily mutually independent..."

In the linear structural equation models (SEM) settings, is independence of \epilson_i required? Otherwise it may violate the markov condition and it seems to be a mixture of directed and undirected graphical model.

(1) line 55: definition of the directed path says that (V_i) is a directed path from V_i to itself. Then, Pa(V_i) should include V_i.

line 91, the conditions in Theorem 1 is not clear. It should clarify that "Suppose we have observed X generated according to the mixing procedure (4) in a number of domains, t = 1, 2, T"

If T =2, then is the considered model is X = Y M \epsilon_1 + (1- Y) M \epsilon_2 where Y is Bernoulli distribution with known/unknown probabilities?, X_j = \beta X_{Pa(j)} + Y_{1j} \epsilon_{j1} + (1 - Y_{1j}) \epsilon_{j2} where (Y_{1j}) are independent Bernoulli distribution with known/unknown probabilities, or X = M (p \epsilon_1 + (1- p) M \epsilon_2)? In addition, in the finite sample setting, is it assumed to be known what distribution of error (\epsilon_t) the sample follows?

line 116, the definition of the bottleneck should be clarified when J and K are sets of nodes.
"We say that a set B is a bottleneck from J to K if every directed path from J to K includes some b \in B."
line 121 From the definition, for each Vi, Ch(Vi) is a bottleneck from Ch(Vi) to X.

First of all, if J includes two nodes, saying J = {1, 2}, and K = {3}, the directed path from J to K is not defined.
It is one of the key definitions to provide an important condition in this paper, it should be precisely defined.

e.g., I at first conjecture that it means that the directed path from j to k, for all j \in J and k \in K. Then, all directed paths are (1,3), (2,3). Hence, {3} is only bottleneck. However, then, it is contradictory to line 121.

line 160, it is also unclear that "Rank(M_K^J) = 0 is a violation of classical faithfulness if there...". It is in fact not true that without considering the error variances of (\epsilon). The statement is true if all error variances are the same, otherwise there is no guarantees that "there is a path from J to K but the path coefficients cancel out so that the net effect of J on K is 0".

In addition, the term "bottleneck faithfulness" might better to be modified, because it is completely based on the graph, the original faithfulness assumption is based on the graph and its probability distribution. Hence, the name "bottleneck faithfulness" might lead to incorrect understanding of the concept of the bottleneck faithfulness.

line 179, it should be clarified that "Let V_k \to V denote the variables whose longest path to X has fewer than k nodes."
For example, 1->2->3<-4. Then, by the definition of V_k, we can see V_1 = {3}, V_2 = {2,3,4}, V_3 = {1,2,3,4}, while the longest path is 1->2->3. Hence, it should be the variables whose paths to X has fewer than k nodes.

Figure 4, it is unclear that how Desc(X_2) = Desc(L_1) in all graphs. For instance, in G_1, Desc(X_2) = (X_2, X_3) and L_1 does not exist.

line 285, it should be clarified that the expectation of the variable corresponding to a node is zero, somewhere in Section 2.

I agree that learning a SEM with latent variables is very difficult, and hence, it is worthy to investigate the necessary and sufficient condition for the identifiability. In addition, it is possible that I do not understand some parts of the paper and I amy not be familiar with the line of works. Howeve, without the concrete defintion of the model and concepts, I am pretty sure that common readers cannot understand the part that I fail to understand. Hence, it should clarify the model the paper focuses and it would be better to provide how to check the proposed conditions are satisfied from finite samples or prior information.

Minor:
line 29 not not
line 82 matrix \mathcal{X}
line 173 equation (8) and does (8)
line 287 d-th domain might be t-th domain

******** Updated Review **********
Thanks for the kind answer. I increase the rate based on the answer. I hope the paper contains the model specified in term of math termonology. In addition, the statements that have potential misunderstanding are clarified.


**Time Spent Reviewing:**

12 - 24 hours

---

> ### Author Response · Authors · 2021-08-10
> **Thank you for taking the time to review the paper write these comments!**
>
> Thank you for taking the time to review the paper write these comments! Some of them are very helpful to improve the manuscript’s readability. Thank you for being candid about your level of experience with this literature and problem setup. We hope your comments and concerns are well addressed below. If there are any remaining doubts or points of unclarity, please kindly let us know.
>
> Q1: “line 41/43/84: It is unclear that "\epsilon is a vector of independent noise variables", "\epsilon_i are uncorrelated within each domain, ...", and "\epsilon_i are not necessarily mutually independent..."
> In the linear structural equation models (SEM) settings, is independence of \epilson_i required? Otherwise it may violate the markov condition and it seems to be a mixture of directed and undirected graphical model.”
>
> A1: In line 41, we give the general claim that the components of $\epsilon$ are mutually independent in the first setting. In lines 42 and 43, we give the more specific conditions in the two settings we consider. In the second setting, $\epsilon_i$ are uncorrelated (and hence independent if they are Gaussian) within each domain, but their variances may change dependently across domains.
>
> In the second setting, the $\epsilon_i$ may be dependent across domains (although we reiterate that they are uncorrelated within each domain). If this is the case, it may be desirable to introduce some high-level background variables explaining why the error variances are dependent across domain. Hence, the Markov condition is not violated. We do not try to model those higher level variables; we focus on the linear causal relations among X and L to explain the measured data.
>
> Q2: “line 55: definition of the directed path says that (V_i) is a directed path from V_i to itself. Then, Pa(V_i) should include V_i.”
> A2: Here we just follow the definitions for Pa(V_i) and path given in standard textbooks like Spirtes et al. [1] or Pearl [2]. The definition of path implies that (V_i) is always a directed path from V_i to itself.
>
> Q3: “line 91, the conditions in Theorem 1 is not clear. It should clarify that "Suppose we have observed X generated according to the mixing procedure (4) in a number of domains, t = 1, 2, T"
> If T =2, then is the considered model is X = Y M \epsilon_1 + (1- Y) M \epsilon_2 where Y is Bernoulli distribution with known/unknown probabilities?, X_j = \beta X_{Pa(j)} + Y_{1j} \epsilon_{j1} + (1 - Y_{1j}) \epsilon_{j2} where (Y_{1j}) are independent Bernoulli distribution with known/unknown probabilities, or X = M (p \epsilon_1 + (1- p) M \epsilon_2)? In addition, in the finite sample setting, is it assumed to be known what distribution of error (\epsilon_t) the sample follows?“
>
> A3: Thank you for your suggestion. Note that in each domain, the data was generated according to the same mixing procedure as given in (4), but that the variances of the noise terms may change from domain to domain. In each domain, the data are not a mixture of multiple linear mixing procedures.
>
> We respectfully fear there may have been some misunderstanding regarding the idea of heterogeneous domains. Theorem 1 assumes that the practitioner has access to multiple domains, and knows which samples came from which domain (but need not know anything about the error distribution). Specifically, the t-th domain consists of iid samples from $X = M_X \epsilon_t$, where $\epsilon_t$ is a vector of uncorrelated random errors. This interpretation is consistent with what is written at lines 34, lines 43-44, and 83-86.
>
> Q4: “line 116, the definition of the bottleneck should be clarified when J and K are sets of nodes. "We say that a set B is a bottleneck from J to K if every directed path from J to K includes some b \in B." line 121 From the definition, for each Vi, Ch(Vi) is a bottleneck from Ch(Vi) to X.
> First of all, if J includes two nodes, saying J = {1, 2}, and K = {3}, the directed path from J to K is not defined. It is one of the key definitions to provide an important condition in this paper, it should be precisely defined.
> e.g., I at first conjecture that it means that the directed path from j to k, for all j \in J and k \in K. Then, all directed paths are (1,3), (2,3). Hence, {3} is only bottleneck. However, then, it is contradictory to line 121.“
>
> A4: We will update the condition to avoid possible confusion: “B is a bottleneck from J to K if for every $j \in J$ and every $k \in K$, every directed path from j to k includes some $b \in B$.” For example, if J = K = {1, 2}, then B is a bottleneck from J to K if and only if $\{1, 2\} \subseteq B$. As this example shows, bottlenecks may include elements of J and/or K.
>
> The second source of confusion indicates a trouble with the choice of terminology “bottleneck.” It might be preferable to understand the word “bottleneck” as something like “barricade” throughout the paper, since “bottleneck” seems to have a misleading connotation. Please observe the distinction among bottleneck, minimal bottleneck, and unique minimal bottleneck given in lines 116-120. In your example, while it is true that {3} is a bottleneck from {1,2} to {3}, {1, 2} is also a bottleneck. Of course, {3} is the unique minimal bottleneck, but not the only possible bottleneck. Hence it is possible for $Ch(V_i)$ to be a bottleneck from $Ch(V_i)$ to X, and so we believe there is no contradiction. If you are unconvinced, please kindly let us know.
>
> Q5: “line 160, it is also unclear that "Rank(M_K^J) = 0 is a violation of classical faithfulness if there...". It is in fact not true that without considering the error variances of (\epsilon). The statement is true if all error variances are the same, otherwise there is no guarantees that "there is a path from J to K but the path coefficients cancel out so that the net effect of J on K is 0".
>
> A5: We have used “classical faithfulness” in the standard sense of “faithfulness” as defined in standard references like Spirtes et al. [1]. If $Rank(M^J_K) = 0$, then $M^J_K = 0$. Thus for every $j \in J$ and every $k \in K$, $V_k$ is marginally independent of $\epsilon_j$, since $V_k = \sum_{i=1}^p M^i_k \epsilon_i = \sum_{i \not\in J} M^i_k \epsilon_i$. Hence classical faithfulness requires that there be no path from $\epsilon_j$ to $V_k$; otherwise there would be an independence relation not encoded in the graph, thus violating faithfulness. Observing that $V_j$ is the only child of $\epsilon_j$, faithfulness indeed requires that there be no path from $V_j$ to $V_k$.
>
> Q6: In addition, the term "bottleneck faithfulness" might better to be modified, because it is completely based on the graph, the original faithfulness assumption is based on the graph and its probability distribution. Hence, the name "bottleneck faithfulness" might lead to incorrect understanding of the concept of the bottleneck faithfulness.”
>
> A6: Bottleneck faithfulness is not just based on the graph; it is a relationship between the graph and the specific rank (and thus properties of the distribution), as stated in lines 156-157. Similarly, classical faithfulness is a relationship between a graph and a distribution (in particular, conditional independence relations implied by the distribution). Both involve relationships between the graph and properties of the distribution.
>
> Q7: “line 179, it should be clarified that "Let V_k \to V denote the variables whose longest path to X has fewer than k nodes." For example, 1->2->3<-4. Then, by the definition of V_k, we can see V_1 = {3}, V_2 = {2,3,4}, V_3 = {1,2,3,4}, while the longest path is 1->2->3. Hence, it should be the variables whose paths to X has fewer than k nodes.”
>
> A7: Assuming that 3 is an observed variable, then your example is consistent with our definition. The natural language description at line 179 is intended to be a simple paraphrase of the formal definitions given in (9) and (10). In light of your comment, to avoid any potential misunderstanding, we can paraphrase: “Let $V_k \subseteq V$ denote the variables such that, for all $X_i \in X$, no path from $V_k$ to $X_i$ has more than k nodes.” (This is consistent with the formal definition in (9) and (10), which is used in the formal proof of Lemma 3.)
>
> Q8: “Figure 4, it is unclear that how Desc(X_2) = Desc(L_1) in all graphs. For instance, in G_1, Desc(X_2) = (X_2, X_3) and L_1 does not exist.”
>
> A8: Thanks for catching this. It should read: $Desc(L_2) \cap \mathcal{X} = Desc(X_1) \cap \mathcal{X}$. We will update this in the paper.
>
> Q9: “line 285, it should be clarified that the expectation of the variable corresponding to a node is zero, somewhere in Section 2.”
>
> A9: Thanks for pointing this out. By default we have centered the data, so that the mean is zero. We will make this explicit in the paper.

---

> > ### Comment · Reviewer_FkNC · 2021-08-14
> > **Updated Review**
> >
> > Thanks for the answer.
> >
> > For Q3, I know the model this paper considers. However I wanted to pointe out that it should be clarified in terms of a math terminology.
> >
> > For Q5, I have again checked the faithfulness assumption. However, based on the settings in the paper, I still cannot understand. In addition, unfortunately, I cannot check Spirtes et al. [1]. Because I cannot provide a counter example, I decrease the level of confidence,
> >
> > For Q6, I do not understand the author's response. As Condition 3 (bottleneck faithfulness) in link 156 - 157, "For every J, K if B is a minimal bottleneck from J to K, then Rank (M_K^J) = |B|", I cannot find the randomness or distribution. The M matrix is fixed, not a random. In contrast, the error \epsilon's are random.
> >
> > I appreciate the response again.

---

> > > ### Author Response · Authors · 2021-08-17
> > > **Thank you for your quick response!**
> > >
> > > Thank you for your quick response! We are especially thankful that you carefully read our response and updated your review comments. Below please find a response to your new comments.
> > >
> > > Q1: “For Q3,I know the model this paper considers. However I wanted to pointe out that it should be clarified in terms of a math terminology.”
> > >
> > > A1: Thank you for your suggestion. To avoid any possible misunderstanding, we will follow your suggestion and change the first sentence of the theorem in the manuscript to: “Suppose we have observed X from multiple domains, $t=1, 2, …, T$, and that each domain is generated according to the mixing procedure described in (4), where $M_X$ is constant across domains...”
> > >
> > > Q2: “For Q5,I have again checked the faithfulness assumption. However, based on the settings in the paper, I still cannot understand. In addition, unfortunately, I cannot check Spirtes et al. [1]. Because I cannot provide a counter example, I decrease the level of confidence,”
> > >
> > > A2: Here please let us give the definition. The relevant definition is “If all and only the conditional independence relations true in [a distribution] P are entailed by the Markov condition applied to [a graph] G, we will say that P and G are faithful to one another.” (Spirtes et al. p.13) In case you don’t have access to Spirtes et al. [1], you may find the definition on page 12 of the paper “Introduction to Causal Inference”, by Spirtes (2010) and published in Journal of Machine Learning Research (https://www.jmlr.org/papers/volume11/spirtes10a/spirtes10a.pdf).
> > >
> > > Q3: “For Q6,I do not understand the author's response. As Condition 3 (bottleneck faithfulness) in link 156 - 157, "For every J, K if B is a minimal bottleneck from J to K, then Rank (M_K^J) = |B|", I cannot find the randomness or distribution. The M matrix is fixed, not a random. In contrast, the error \epsilon's are random.”
> > >
> > > A3: First, we completely agree that M depends on the parameters involved in the causal process and is not random.
> > > Second, we wish to emphasize that the original faithfulness assumption (see the definition in response A2) can be seen as constraints on the parameters involved in the causal model. For example, let’s consider the following linear causal model with three variables F, G, and H: G = a*F + E1; H = b*F + c*G  + E2. In other words, the causal structure has three direct causal links: F -> G, F -> H, and G -> H, with linear coefficients a, b, and c, respectively.  Then if ac + b = 0, F and H will be statistically independent, although there is a causal link from F to H.  That is, this independence is not entailed by the Markov condition applied to the causal graph and, accordingly, it is a violation of the faithfulness assumption.
> > > Third, as you pointed out, our bottleneck faithfulness is also a constraint on the parameters involved in the causal process. Although this constraint is different from those required by classical faithfulness, bottleneck faithfulness is in line with classical faithfulness in the sense that both of them impose constraints on the parameters of the causal model.
> > > If you don’t find the explanation clear, please kindly let us know. Once again, we appreciate your nice feedback and further questions.

---

### Official Review · Reviewer_NMAx · 2021-07-11

**Rating:** 8
**Confidence:** 4

**Summary:**

This paper is about statistical causal inference. More specifically, this paper is about causal discovery that infers causal graphs. The data used is observational data without intervention. All the variables are continuous.

This paper considers linear causal models with hidden variables. This setup can be written using an overcomplete mixing matrix as in (1). This overcomplete mixing matrix is known to identifiable up to the permutation and signs of the columns if the hidden variables are non-Gaussian and independent [4]. This is known in the field of independent component analysis. When the number o the hidden variables are the same as the observed variables, it is known that changing noise variance assumption is sufficient to make the overcomplete mixing matrix identifiable even when the hidden variables are Gaussian [16]. A contribution of this paper is Theorem 1 that extends the identifiability result [16] to overcomplete cases.

Based on the two identifiability results, they further consider the sufficient and necessary conditions of the linear causal models after the overcomplete mixing matrix has been identified. They aim to identify the underlying causal model beyond the Markov equivalence class.

They further assume the sparseness of the causal adjacency matrix F, i.e., the sparsest causal model is true if multiple causal models give the same overcomplete mixing matrix M.


**Limitations And Societal Impact:**

More discussion on the assumption that the sparsest graph is the truth would be desirable.

**Main Review:**

Originality: This paper tries to investigate the sufficient and necessary conditions for linear causal models with hidden variables, including linear causal models with hidden common causes and those with latent factors being causally related. This general framework would be new because those models have been investigated separately.

The estimation method based on exhaustive search in Section 3 looks a naïve method. Another proposed estimation based on sparse regularization is proposed. The idea seems not so new.

Quality:
I’m not sure if the assumption on the sparseness on lines 106-111 is reasonable or effective in which application domains. I wonder if there is a reasonable scientific reason to think that the sparsest models are the truth.

Further, I’m interested in how the identifiability results change if the sparsest model assumption has been removed. I thought previous works [17-22] do not make this assumption.

I would appreciate it if the authors make these points above clear.

The proposed estimation methods look not very fresh or very solid. One is a simple exhaustive search. It is not clear if the other regularization method is consistent or consistently selects the truth. In fact, they say this regularization method did not work well (lines 317-324).

I was not sure if an underdetermined version of [16] has been already considered in the literature of independent component analysis or signal processing.

Clarity: There are serval unclear things.
1.	Line 116: A set B. Is B a set of what?
2.	Line 29: might not not - > might not
3.	Line 133: Bottleneck faithfulness
Bottleneck faithfulness has not been defined at this point.
Figure 2: “is not a redundancy” What would be the definition of this?
Figure 3: Co-parental non-redundancy, parental non-redundancy
I couldn’t find their definitions in the paper. Ah, I see. I found them in the supplement.
4.	Line 192: at least one?
What is this ? here?
5.	In the supplement, on page 11, what Times of Times (i) means?
6.	In 6.1, were the artificial data generated under the assumption that the underlying model is the sparsest.
7.	How is the number of hidden variables selected?

Significance:
This work could contribute to a deeper theoretical understanding of causal discovery methods in the presence of hidden variables, including hidden common causes and latent factors, by considering a general framework includes both classes of linear causal models with hidden common causes and latent factors.
Relating to this, I have a question. Previous works [17-22] gave sufficient conditions. Did the authors of this submission find some other sufficient conditions for any of the models considered in [17-22]? If there are no other sufficient conditions, does this mean that the conditions given by [17-22] turned out to be also necessary?

**Time Spent Reviewing:**

20

---

> ### Author Response · Authors · 2021-08-10
> **Thank you for your valuable and detailed comments!**
>
> Thank you for your valuable and detailed comments!  We have collected them into three major remarks and seven minor questions. We are excited that the reviewer has hit on two major points that need a bit more motivation in the manuscript. If there are still doubts about these responses, please kindly let us know.
>
> Q1: “The proposed estimation methods look not very fresh or solid.”
>
> A1: We agree that both of our estimation methods are standard: L1 regularization and BIC score. However, we see the main contribution of our paper as a theoretical one. Existing work on this problem has progressed inefficiently because it is generally not clear which partially observed structures are identifiable, and which are not. Hence a significant part of the process involves coming up with a type of identifiable graph, proving its identifiability, and then designing an efficient estimation algorithm. Our conditions will hopefully inspire researchers to jump straight to step three.
>
> With that said, we have included experimental results of an exhaustive search method as a sort of proof of concept. It shows that with finite samples, we can reliably identify the true causal structure if and only if the causal structure satisfies the bottleneck condition, strong non-redundancy, and bottleneck faithfulness. This serves as a sanity check for our identification theory, and also gives us some idea what the sample complexity looks like. We have further included our discouraging results from the L1 method to caution practitioners against such a simple but flawed approach, and to show readers that the estimation problem remains open. We do not advocate its application as an estimation method, but hope for more advanced estimation methods in light of our theoretical results.
>
> Q2: “Further, I’m interested in how the identifiability results change if the sparsest model assumption has been removed. I thought previous works [17-22] do not make this assumption.”
>
> A2: Thank you for this interesting question. We believe that references [19-22] actually do assume minimality implicitly; that Salehkaleybar et al. [18] is only solving a subset of our problem (and therefore has no need to assume minimality); and that Hoyer et al. [17] take an even less plausible primitive than minimality: namely, that no two latent variables are adjacent (more precisely, they translate graphs with latent-latent interactions into “canonical models” with no latent-latent interactions, and then learn these false graphs).
>
> Each of reference [19, 20, 21, 22] assumes that each latent variable has at least four [19], three [20], or two [21, 22] pure observed children (observed variables which have no other parents). As we mention at lines 234-239, these assumptions are strictly stronger than our bottleneck and non-redundancy conditions. [The proof is straightforward: Non-redundancy clearly holds because for every $L_i$ and $L_j$, $L_i$ has at least two children that $L_j$ does not. The bottleneck condition clearly holds because if, for some $L_i$, some latent child $L_j$ is not included in a bottleneck from $Ch(L_i)$ to X, then $L_j$’s two (or more) pure children would have to be included.] Since these two conditions along with bottleneck faithfulness (which, as we have shown in supplement section 1.3, is generally applicable) are sufficient to prove that the graph is uniquely minimal (see Theorem 3 and the definition of “identifiable up to trivialities” at 109-111) it follows that references [19, 20, 21, 22] all assume conditions strictly stronger than minimality.
>
> In particular, none of references [19, 20, 21, 22] can identify the structures in Figures 2, Figure 4, and (more generally) any structure where no latent variable has any pure children. However, we can identify any graph that those papers can.
>
> While Salehkaleybar et al. [18] do not assume minimality, they are only trying to solve a part of the problem that we are. As we remark in line 229, they want to identify the net effect of each observed variable on every other observed variable. Answering that question does not require the them to single out any particular latent structure.
> As stated in line 225, the discussion of “canonical models” in Hoyer et al. [17] can be seen as a way of translating models which violate our bottleneck condition (and are therefore not uniquely minimal) into models which do satisfy our bottleneck condition. These "canonical models" assume that the latent variables are mutually independent (and accordingly, not causally adjacent). In other words, they don't try to learn causal graphs with causally adjacent latent variables; they take as primitive that the latent variables are nonadjacent. But we try to go further; we study the identifiability of causal models with causally adjacent latent variables. So we need to take a different primitive: a particular type of minimality.
>
> Q3: “I’m not sure if the assumption on the sparseness on lines 106-111 is reasonable or effective in which application domains. I wonder if there is a reasonable scientific reason to think that the sparsest models are the truth.”
>
> A3: Thank you for raising this concern. First, let’s summarize the logical structure of the argument appearing in the paper. For any partially observed causal structure, it is always possible to construct a denser model which yields the same distribution, as we indicate in lines 100-101. This is a huge equivalence class; if we are to even talk about identification theory for partially observed causal models, we need a way to move beyond this huge equivalence class. This motivates us at in lines 106-111 to chose the simplest model in the sense that we select from this class the graph or graphs with the smallest number of edges. In this sense, we don’t assume minimality so much as we take it as an foundational principle: roughly speaking (see lines 109-111) to be identifiable is to be minimal. The bottleneck, non-redundancy, and bottleneck faithfulness conditions provide an interpretable and (given the graph) testable characterization of the class of minimal graphs (which, as we have argued here and in lines 100-111, are exactly the identifiable graphs).
>
> Variants of this minimality principal are common throughout the causal discovery literature. In our response to Q2, we have argued that the assumptions of references [19-22] all have unique minimality as a logical consequence. But similar principles are widely adopted in causal discovery even in the causally sufficient case. For example, Forster et al. (2018) introduce a principle they call “frugality” in the causally sufficient case, and argue that such a principle has several desirable properties. Raskutti and Uhler (2012) similarly propose a principle they call the “sparsest Markov representation” assumption. More generally, from a model selection perspective, popular model selection criteria like BIC favor minimal graphs, as we show in our experiments.
>
> Seven Minor Questions:
>
> Also, thanks for your seven minor questions. We will clarify these in the manuscript. Our clarifications:
> 1) We should say, “Let J, B, and K be subsets of nodes in a directed graph. Note that they need not be mutually disjoint.”
> 2-4) Thanks for catching these typos.
> 5) it means “the number of trials (out of 10) that graph (i) was the best fitting,” as indicated in the paragraph introducing these tables.
> 6) Q: “In 6.1, were the artificial data generated under the assumption that the underlying model is the sparsest.”
> A: We simulated data from both identifiable (and therefore “sparsest,” or in the terminology of the paper, “minimal”) and non-identifiable (including both minimal and non-minimal) graphs. This is detailed in lines 290-313 of the main paper.
> 7) Q: “How is the number of hidden variables selected?”
> A: In the exhaustive search experiment, the BIC score naturally selects the number of hidden variables. As indicated in line 294, all graphs with 3 observed variables were enumerated in order from the sparsest to the densest, subject to the constraint that every latent have at least two neighbors (children or parents) and at least one observed descendant. (The constraint just ensures that every latent contributes to the model in some way.) For the L1 experiment, we simply used the true number of latents; the point is to show that the $L_1$ penalty is not effective, and that more sophisticated estimation procedures are needed. Such methods are complicated enough to require separate studies their own; thus we restrict this paper to identification theory and some stimulating proof of concept.
>
> References
> M. Forster, G. Raskutti, R. Stern, and N Weinberger. The Frugal Inference of Causal Relations. The British Journal for the Philosophy of Science 69(3), 2018
> G. Raskutti and C. Uhler. Learning directed acyclic graph models basedon sparsest permutations. Stat 7(1), 2018.

---

> > ### Comment · Reviewer_NMAx · 2021-08-19
> > **On A2 and A3**
> >
> > Thanks a lot for your feedback!
> >
> > Let me make sure of my understanding. So, does your paper propose another "sparse" model and consider its sufficient and necessary conditions rather than generalize the previous works on models with latent factors or latent common causes in the references? Further, you are saying that each of the previous works proposed their own "sparse" models?

---

> > > ### Author Response · Authors · 2021-08-20
> > > **Many thanks for your questions!**
> > >
> > > Many thanks for your questions! Please find our response below. Throughout this response, we will always use the word “sparsity” with the same formal meaning as identifiability up to trivialities, defined at line 110 of our paper. The reader is free to instead read it as minimality, defined formally in line 107 of our paper; identifiability up to trivialities is a special type of “unique” minimality, so the entailment claims will still hold.
> > >
> > > Q1: [D]oes your paper propose another "sparse" model and consider its sufficient and necessary conditions rather than generalize the previous works on models with latent factors or latent common causes in the references?
> > >
> > > A1: Previous works [19, 20, 21, 22] provide conditions which entail sparsity (with precise meaning defined above). Our conditions also entail sparsity (again, as defined above). However, previous conditions are clearly much more restrictive than ours; our conditions are strictly weaker than any of the previous conditions. In particular, existing works cannot recover the structures in Figures 2, Figure 4, or (more generally) any structure where no latent variable has any pure children. Our conditions can.
> > >
> > > Q2: Further, you are saying that each of the previous works proposed their own "sparse" models?
> > >
> > > A2: Each of references [19, 20, 21, 22] takes sparsity as a primitive; each then provides graphical conditions which entail sparsity (as defined in the first paragraph above). We take the same primitive, and provide graphical conditions which not only entail sparsity (again, as defined above), but which are also necessary for identifiability up to trivialities (which, we recall, is a special type of “unique” minimality).
> > >
> > > Hoyer et al. [17] takes a different primitive: the latent factors are independent, so that no latent factor has any parent (whether latent or observed). Such a primitive clearly makes it impossible to identify more interesting structures in which latent causes interact with one another.

---

> > > > ### Comment · Reviewer_NMAx · 2021-08-20
> > > > **I have some more questions.**
> > > >
> > > > Thanks a lot for your reply! I want to ask some more questions to understand the submission better.
> > > >
> > > > 1. Can your method do all of what non-Gaussian models with latent factors in [20,21,22] can do?
> > > >
> > > > 2. You said that Hoyer et al. (2008) (and probably Salehkaleybar et al., 2020) consider canonical models and estimate false graphs with independent hidden common causes. But, I thought their canonical models still give correct causal orderings of observed variables, and they don't aim to estimate causal relations of latent variables. Though this is a bit similar question above, can your method do all of what non-Gaussian models with latent common causes in Hoyer et al. (2008) and Salehlakeybar et al. (2020) can do? I wonder if this may not be so since your method is trying to estimate both observed and latent variables' causal relations. So, is your assumption weaker than theirs, or are they not inclusive to each other?

---

> > > > > ### Author Response · Authors · 2021-08-23
> > > > > **We have some more responses.**
> > > > >
> > > > > Many thanks for the follow-up questions! Let us take the opportunity to provide detailed responses.
> > > > >
> > > > > Q1. Can your method do all of what non-Gaussian models with latent factors in [20,21,22] can do?
> > > > >
> > > > > A1. Yes, our method can do everything [20, 21, 22] can do and more.
> > > > >
> > > > > Intuitively, it is because they tried to provide sufficient conditions for the identifiability of the latent variables and their relations, while our work aims to provide necessary and sufficient conditions (which are strictly weaker, as seen from the argument in lines 234-239 and the examples we provided in A1 of our previous reply). In addition to having these more restrictive conditions on the latent part of the structure, references [20, 21, 22] do not allow any observed variable to be a cause of any other (latent or observed) variable. Our conditions do allow observed variables to have children.
> > > > >
> > > > > Q2. You said that Hoyer et al. (2008) (and probably Salehkaleybar et al., 2020) consider canonical models and estimate false graphs with independent hidden common causes. But, I thought their canonical models still give correct causal orderings of observed variables, and they don't aim to estimate causal relations of latent variables. Though this is a bit similar question above, can your method do all of what non-Gaussian models with latent common causes in Hoyer et al. (2008) and Salehkaleybar et al. (2020) can do? I wonder if this may not be so since your method is trying to estimate both observed and latent variables' causal relations. So, is your assumption weaker than theirs, or are they not inclusive to each other?
> > > > >
> > > > > A2. This is a very interesting question, since Hoyer et al. [17] (hereafter HSKP), Salehkaleybar et al. [18] (hereafter SGKZ), and our submission each consider very different problem setups. We aim to identify the whole causal structure, including 1) the causal adjacencies among latent variables, 2) those between latent variables and observed variables, and 3) those among observed variables. Both HSKP and SGKZ were concerned with various aspects of 3), such as the causal order. Before diving into a deep comparison of HSKP, SGKZ, and our own contribution, let us emphasize a few points. First, we reiterate that our conditions are weaker than any previous conditions that identify the full causal structure among both latent and observed variables -- i.e. that simultaneously identify 1), 2) and 3). Second, since HSKP and SGKZ only identify relatively easy aspects of 3), it is natural that our conditions are not strictly weaker than theirs. But, our conditions are not strictly stronger, either. In fact, our novel conditions point towards an improved version of Theorem 16 from SGKZ; thus for the specific task considered in that theorem, we are able to do everything SGKZ can while requiring less restrictive assumptions than SGKZ. Finally, both HSKP and SGKZ assume classical faithfulness among observed variables, which promotes the selection of sparser graphs. Please find our detailed analysis below.
> > > > >
> > > > > One of the questions considered in SGKZ is, “Under what conditions is the causal order among observed variables identifiable in a partially observed linear non-Gaussian causal model?” In Lemma 5, they show that if $\mathcal{M}$ (defined in (5) of our paper) is identified, then the causal order is identified if classical faithfulness holds:
> > > > >
> > > > > Assumption 1 (Classical Faithfulness): If there is a path from $X_i$ to $X_j$, then $M^i_j \neq 0$.
> > > > >
> > > > > This condition is of course strictly weaker than ours; as we write in lines 160-163, bottleneck faithfulness entails classical faithfulness. But, the task is also strictly easier than ours, in the sense that if F is identified up to trivialities then the causal order among observed variables is also identified (while the causal order alone tells very little about F). If the practitioner only cares about the causal order among observed variables, then there is no reason to require anything more than classical faithfulness. But if the practitioner is further interested in the more difficult problems we consider in our paper, such as learning the latent causal structure or learning the mechanism by which one observed variable influences another, then additional assumptions are clearly necessary.
> > > > >
> > > > > Remark: Notice that classical faithfulness can be understood as a preference for causal graphs with as few causal edges as possible. Explicitly, it is easy to see that classical faithfulness minimizes the number of pairs of observed variables for which one has a path to the other. This is yet another example of a minimality-type assumption in previous causality research.
> > > > >
> > > > > HSKP considers a related task. They aim to identify an equivalence class of the possible adjacency matrices among observed variables. Notice that they are not looking for a unique adjacency matrix; they are looking for an equivalence class, and the matrices in this class might not all imply the same causal graph among observed variables. Notice also that their conditions are not strictly weaker than ours: they do not allow latent mediators, but we do. (This is indeed an additional assumption on top of canonical form.) Even with this relatively strong assumption that there are no latent mediators, the resulting equivalence class is still huge. While HSKP shrinks this class somewhat by applying the classical faithfulness assumption, they do not arrive at any theory of unique identification, even in this very restricted setting. In particular, under their assumptions even the observed part of the causal graph is not always identifiable. In summary, we recover more information (uniquely identifying the observed variables’ adjacencies) but our conditions are neither stronger nor weaker.
> > > > >
> > > > > To move beyond this equivalence class and identify the observed part of the causal graph uniquely, additional assumptions are needed. One assumption sufficient for this purpose is given by SGKZ:
> > > > >
> > > > > Assumption 2: For any observed $X_i$ and latent $L_j$, $Desc(X_i) \cap \mathcal{X} \neq Desc(L_j) \cap \mathcal{X}$.
> > > > >
> > > > > Under assumptions 1 and 2, Theorem 16 (SGKZ) shows that $M^X_X$ is identified uniquely given $\mathcal{M}$. Clearly, if we make HSKP’s further assumption that there are no latent mediators, $F^X_X$ is identified uniquely from $M^X_X$. However, we recall that this is still quite a restrictive assumption compared to our modeling assumptions.
> > > > >
> > > > > Furthermore, our paper implies that Assumption 2 is not necessary. As we mention in lines 231-233, Assumption 2 entails a weakened version of non-redundancy:
> > > > >
> > > > > Assumption 2.1 (Weak non-redundancy): For all $L_i$ and $X_j$, $Ch(L_i) \not\subseteq Ch(X_j) \cup \{X_j\}$.
> > > > >
> > > > > However, Assumption 2.1 does not entail Assumption 2.
> > > > >
> > > > > This weakened version of non-redundancy (along with bottleneck faithfulness, which is a very mild assumption as we argue in Section 1.3 of the supplement) is actually necessary and sufficient to solve this task. (The proof is long, but straightforward, and closely follows the proofs of Theorem 2 and Lemmas 2 and 3 of our paper.) Thus while we do not consider this particular task (as it is strictly easier than the one we consider), our graphical conditions naturally lead to an improved version of Theorem 16 from SGKZ.
> > > > >
> > > > > To summarize, HSKP and SGKZ consider several related problems, all of which are strictly easier than the main problem we consider in our own paper. One of them, the causal order of the observed variables in a partially observed linear non-Gaussian model, requires only classical faithfulness. We do not pretend to improve on this solution. A second problem involves identifying the direct effect of one observed variable on another. HSKP is only able to identify an equivalence class of possibilities, and requires that there be no latent mediators. This solution is less informative than ours, and the model assumptions are neither stronger nor weaker than ours. A third problem involves identifying the net effect of one observed variable on another. While we do not consider this problem in our paper, our conditions can naturally be adapted to necessary and sufficient conditions for this problem; in particular, this is an improvement on the conditions presented by SGKZ.
> > > > >
> > > > > Once again, thank you for this very interesting question!

---

> > > > > > ### Comment · Reviewer_NMAx · 2021-08-24
> > > > > > **Thanks a lot for your feedback!**
> > > > > >
> > > > > > I have some more questions.
> > > > > >
> > > > > > 1. I didn't think that HSKP assumes no hidden intermediate variables. See Fig. 2 of HSKP. I thought that if the relations are linear, even if omitting intermediate variables, their model is still linear, and this does not affect the causal orders of variables.
> > > > > >
> > > > > > 2. *  both HSKP and SGKZ assume classical faithfulness among observed variables, which promotes the selection of sparser graphs.
> > > > > >
> > > > > > I didn't follow how classical faithfulness is related to sparseness. I wonder if assuming faithfulness would not necessarily mean assuming some sparseness. For example, this paper seems to consider the classic PC algorithm with faithfulness and FURTHER takes sparse graphs.
> > > > > > https://www.jmlr.org/papers/volume8/kalisch07a/kalisch07a.pdf
> > > > > >
> > > > > > 3. I'm interested in whether model candidates considered in SGKZ could be different from those in this submission. I wonder if the minimal assumption in lines 106-108 might lead to considering sparser graphs than those SGKZ.
> > > > > >
> > > > > > 4. * A1. Yes, our method can do everything [20, 21, 22] can do and more.
> > > > > >
> > > > > > Ok. I understand your method tries to further find causal relations of observed variables and those of observed variables and latent factors in addition to the causal relations of latent factors as  [20, 21, 22] do. Then, I'm interested in whether arbitrary DAG structures are allowed between observed variables, under your assumptions and maybe under the same assumptions on the causal relations of latent factors made in [20, 21, 22], e.g., not having L1 in Fig 2 of this submission.
> > > > > >
> > > > > > I would like to know how users should use different methods by asking these questions.

---

> > > > > > > ### Author Response · Authors · 2021-08-25
> > > > > > > **Thank you for your questions!**
> > > > > > >
> > > > > > > Thank you for your questions and your consideration of our responses!
> > > > > > >
> > > > > > > Q1. I didn't think that HSKP assumes no hidden intermediate variables. See Fig. 2 of HSKP. I thought that if the relations are linear, even if omitting intermediate variables, their model is still linear, and this does not affect the causal orders of variables.
> > > > > > >
> > > > > > > A1. By “latent mediator” or “hidden intermediate variable,” we understand a latent variable which has at least one observed ancestor and at least one observed descendant. We agree that ignoring this variable does not change the causal order among observed variables. If the practitioner only wants to learn the causal order among observed variables, they should make use of Lemma 5 in SGKZ and make the very mild assumption of classical faithfulness, as discussed in Q2 of our response provided on August 23.
> > > > > > >
> > > > > > > We agree that the work by HSKP can handle intermediate confounders in the sense that it can recover the canonical form corresponding to a true model with intermediate confounders. As a consequence, the causal order of the observed variables may be recovered. But because no latent variable in a canonical form has any parents, HSKP cannot identify whether any of the latent variables are mediators.
> > > > > > >
> > > > > > > Q2. I didn't follow how classical faithfulness is related to sparseness. I wonder if assuming faithfulness would not necessarily mean assuming some sparseness.
> > > > > > >
> > > > > > > A2. For causal graphs with no latent variables, classical faithfulness entails edge minimality (or frugality), which is a particular type of sparseness condition (see, e.g., Section 3 of “The Frugal Inference of Causal Relations” by Forster et al., 2018).
> > > > > > >
> > > > > > > In the partially observed case, because we have only measured some of the variables, it depends on how one defines faithfulness (e.g., would classical faithfulness apply to confounders as well?) and sparseness (e.g., is it the sparseness of the whole graph, or just the edges among observed variables?). Because we want to make the least restrictive assumptions possible for the task at hand, it is the task that motivates which of these definitions we choose. Accordingly, for our task, we assume bottleneck faithfulness, which involves the latent variables and (as discussed in Section 4.1 of our paper) helps to promote minimality over the full graph when estimating the full graph (including latent variables).
> > > > > > >
> > > > > > > Q3. I'm interested in whether model candidates considered in SGKZ could be different from those in this submission. I wonder if the minimal assumption in lines 106-108 might lead to considering sparser graphs than those SGKZ.
> > > > > > >
> > > > > > > A3. First, according to our understanding,SGKZ does not try to identify any graphs; it only tries to identify the observed-observed component of the mixing matrix, $M^X_X$.
> > > > > > >
> > > > > > > However, our previous reply’s analysis of Theorem 16 applies to all graphs, sparse or dense. We summarize that analysis here. Our conditions suggest an improvement to Theorem 16 of SGKZ. If the practitioner is willing to make Assumption 2, then they should be even more willing to replace it with the strictly less restrictive Assumption 2.1 (Weak Non-redundancy). If $M^X_X$ is identifiable by SGKZ, then it is also identifiable under our bottleneck faithfulness and weak non-redundancy conditions; however, the converse is often false.
> > > > > > >
> > > > > > > Q4. I understand your method tries to further find causal relations of observed variables and those of observed variables and latent factors in addition to the causal relations of latent factors as [20, 21, 22] do.
> > > > > > >
> > > > > > > A4. Yes, our paper has the added benefit of identifying the children of observed variables. But that’s not all. Even if there is only very simple confounding, we can identify structures that [20, 21, 22] cannot.
> > > > > > >
> > > > > > > For example, consider the graph $X_1 \gets L_1 \to X_2 \gets L_2 \to X_3$. Here there are no latent-latent interactions, nor any observed-observed interactions, nor any latent mediation. Nevertheless, this graph violates the assumptions of each of references [20, 21, 22], because each latent variable has only one pure child. ($X_2$ is not pure because it has two parents.) By contrast, this graph does satisfy our conditions. (The bottleneck condition is satisfied trivially, and non-redundancy is satisfied because each of $L_1$ and $L_2$ has a child the other does not.)
> > > > > > >
> > > > > > > Again, our paper has the added benefit of identifying the children of observed variables. But this example shows that even if there is only very simple confounding, we can identify structures that [20, 21, 22] cannot.
> > > > > > >
> > > > > > > Q5. Then, I'm interested in whether arbitrary DAG structures are allowed between observed variables, under your assumptions and maybe under the same assumptions on the causal relations of latent factors made in [20, 21, 22], e.g., not having L1 in Fig 2 of this submission.
> > > > > > >
> > > > > > > A5. In our model, DAG structures among observed variables are restricted by non-redundancy. In particular, for all $L_i$ and $X_j$, $Ch(L_i) \not\subseteq Ch(X_j) \cup \{X_j\}$. Other than this, we place no restrictions on the observed-observed adjacencies. By contrast, References [20, 21, 22] do not allow any observed-observed interactions whatsoever.
> > > > > > >
> > > > > > > The question regarding Figure 2 is handled in A4 above.
> > > > > > >
> > > > > > > Q6. I would like to know how users should use different methods by asking these questions.
> > > > > > >
> > > > > > > A6. There are many practical issues to consider about a partially observed linear causal system in order to solve real problems. In general, harder tasks (such as estimating the whole causal graph including latent confounders and their relations) tend to require stronger assumptions; easier tasks (such as identifying the causal order of only the measured variables) tend to require weaker assumptions. We advocate making the weakest assumptions needed for the task at hand. It is for this reason that our paper gives conditions which are jointly necessary and sufficient to identify the entire adjacency matrix up to trivialities; they cannot be made weaker if F is to be identified.
> > > > > > >
> > > > > > > If the practitioner does not want to recover all of F, but only certain properties of the partially observed linear system, then the latter (easier) task will not require such strict assumptions. For instance, see HSKP and SGKZ (as well as A2 in the response provided on August 23) for their tasks and sufficient conditions.
> > > > > > >
> > > > > > > Once again, we appreciate your questions and your consideration of our response.

---

> > > > > > > > ### Comment · Reviewer_NMAx · 2021-08-27
> > > > > > > > **Model class**
> > > > > > > >
> > > > > > > > Thanks a lot for your feedback. It helped me a lot to understand the relations of your work with the previous works.
> > > > > > > >
> > > > > > > > I'm interested in how enough "rich" structures that are useful in practice are allowed. For example, is there any possibility that your assumptions including the minimality and bottleneck faithfulness etc. somehow prefer some particular relations among multiple models included in M, e.g., they might prefer directed edges btw observed variables than those btw latent factors or the reverse?

---

> > > > > > > > > ### Author Response · Authors · 2021-08-29
> > > > > > > > > **Thank you for your continued interest!**
> > > > > > > > >
> > > > > > > > > We are grateful for your continued interest in the paper. The “richness” of allowed structures is restricted only by lines 106-110: when selecting a partially observed graph to explain the observed overcomplete basis $\mathcal{M}$, we always prefer sparser graphs (counting all edges equally) over denser graphs (counting all edges equally). There are several reasons to count all edges equally; for instance, from the perspective of model selection based on the number of model parameters, every edge contributes equally to the total number of free parameters in the model.
> > > > > > > > >
> > > > > > > > > Of course, it is quite difficult to know which graphs satisfy the minimality condition of the preceding paragraph -- in the language of the paper, it is hard to know what graphs are identified up to trivialities. For this reason, we have presented the bottleneck and strong non-redundancy conditions, which (assuming bottleneck faithfulness) are logically equivalent to (6) due to Theorems 2 and 3. These two conditions are much easier to check. For some consequences of these conditions, please see A2 of our response to reviewer jP5B.
> > > > > > > > >
> > > > > > > > > Please let us know if there are other questions.

---

> > > > > > > > > > ### Comment · Reviewer_NMAx · 2021-08-29
> > > > > > > > > > **Thanks again for your feedback!**
> > > > > > > > > >
> > > > > > > > > > Thanks again for your feedback! I just wondered that if we want to explain the association btw two of the observed variables, adding a directed edge between the two observed variables would give a sparser graph than adding directed edges from a hidden common cause to the two because the former adds one directed edge and the latter adds two directed edges. But, of course, things can be more complicated.

---

> > > > > > > > > > > ### Author Response · Authors · 2021-09-02
> > > > > > > > > > > **Thank you for the clarification!**
> > > > > > > > > > >
> > > > > > > > > > > Thank you for the clarification! We agree that in your example, the first graph is sparser than the latter. However, these two systems have different mixing matrices, which are identifiable in the heterogeneous or non-Gaussian case. Hence in the linear heterogeneous or linear non-Gaussian case, the graphs have different population distributions and are not observationally equivalent. Our minimality principle commits us to choosing the sparsest graph that generates the population distribution -- not merely the sparsest graph.
> > > > > > > > > > >
> > > > > > > > > > > In contrast, if we were to consider the partially observed linear Gaussian case and had only two observed variables, the two graphs would be observationally equivalent. The identifiability conditions of the Gaussian case are different from those of the non-Gaussian or heterogeneous case. However, as indicated by the title, the Gaussian case is not studied in this paper.

---

> > > > > > > > > > > > ### Comment · Reviewer_NMAx · 2021-09-02
> > > > > > > > > > > > **Some more question**
> > > > > > > > > > > >
> > > > > > > > > > > > Though that was not a good example, I was trying to clarify my point. After all, my point would be something like this:
> > > > > > > > > > > > Your minimality assumption would help pick a single causal graph, but I'm a bit worried if the sparsest graph might be very different from the correct one. Is there any common structure between such causal graphs that have the same mixing matrix? (For example, the graphs in the same Markov equivalence class share the same skeleton for DAG cases with no hidden variables. I understand that your work is trying to do more complicated things. I just mentioned the Markov equivalence class to explain my question.)

---

> > > > > > > > > > > > > ### Author Response · Authors · 2021-09-02
> > > > > > > > > > > > > **Response to clarified clarification**
> > > > > > > > > > > > >
> > > > > > > > > > > > > Thanks for the additional clarification! The question of equivalence classes is very interesting. Some insight about these equivalence classes can be gleaned by a careful inspection of the proof of Theorem 2 and Figure 6 in Section 2.2 of the supplement. We’ll try and gather this intuition in our response. To begin, it’ll help to introduce some terminology.
> > > > > > > > > > > > >
> > > > > > > > > > > > > As in the paper, $X_i$ corresponds to an observed variable, $L_i$ to a latent variable, and $V_i$ to either kind of variable. We’ll say that $V_i$ violates the bottleneck condition weakly if $Ch(V_i)$ is a minimal bottleneck but not uniquely minimal (those definitions are given in lines 116-120), and strongly if $Ch(V_i)$ is not even a minimal bottleneck. We’ll call $L_i$ a redundancy of $V_j$ if $Ch(L_i) \subseteq Ch(V_j) \cup V_j$. We’ll call a redundancy parental if $L_i \to V_j$, and co-parental otherwise. (Note that redundancies can’t have $L_i \gets V_j$ by definition.) We will use $Ch'(V_i)$ to denote the children of $V_i$ according to a newly constructed graph.
> > > > > > > > > > > > >
> > > > > > > > > > > > > With that out of the way, there are two cases to consider here. First, if the adjacency matrix is not minimal (in the precise sense of line 107), then (as far as we are aware) there is no existing method that will allow us to recover the true latent structure uniquely. In particular, none of references [19-23] will identify the true latent structure. In order to identify a non-minimal graph uniquely, the identification theory would have to be built around some starting principle other than minimality -- and hence, other than the number of free parameters in the model.
> > > > > > > > > > > > >
> > > > > > > > > > > > > One way that a graph can be non-minimal is if  there is an strong bottleneck violation. For an example of such a graph and its minimal counterpart, see Figure 1. It is clear from the proof of Theorem 2 that at least one equivalent minimal graph will have $Ch'(V_i) \subseteq Desc(V_i)$ for every $V_i$. Another way a graph can be non-minimal is if there is a violation of bottleneck faithfulness, as shown in Section 4.1 of the main paper; however, Section 1.3 of the supplement shows that this never happens in practice.
> > > > > > > > > > > > >
> > > > > > > > > > > > > If the goal is to construct the set of all (not necessarily minimal) graphs that are observationally equivalent to a particular graph, the problem becomes rather straightforward. For example, the algorithm could look something like this:
> > > > > > > > > > > > >
> > > > > > > > > > > > > 1) Calculate the overcomplete basis $\mathcal{M}$ according to (3) and (5).
> > > > > > > > > > > > >
> > > > > > > > > > > > > 2) For every permutation of the columns of $\mathcal{M}$, do steps a) through e):
> > > > > > > > > > > > >
> > > > > > > > > > > > > a) if there is a pair $(i, j)$ with $i < j$ and $Support(M^i) \subseteq Support(M^j)$, go on to the next permutation.
> > > > > > > > > > > > >
> > > > > > > > > > > > > b) for every $i$, if $M^i_j \neq 0$ and $M^k_j = 0$ for every $k > i$, then associate $M^i$ with $X_j$.
> > > > > > > > > > > > >
> > > > > > > > > > > > > c) put $v_i = M^i – M^i_j I^j$ if $M^i$ is associated with $X_j$ and $v_i = M^i$ otherwise. (Recall that $I^j$ is one-hot in the $j$-th slot.)
> > > > > > > > > > > > >
> > > > > > > > > > > > > d) collect $\mathcal{J}(i) = \{J \subseteq \{i+1, i+2, …, p\} : v_i \in Range(M^J) \}$.
> > > > > > > > > > > > >
> > > > > > > > > > > > > e) For each $(Ch'(V_1), …, Ch'(V_p)) \in \mathcal{J}(1) \times … \times \mathcal{J}(p)$, add the corresponding graph to the equivalence class.
> > > > > > > > > > > > >
> > > > > > > > > > > > > This algorithm closely follows the intuition of the proof of Theorem 3; step a) plays a similar role as Lemma 3, steps b-c) play a similar role as Lemma 4, and steps d-e) play a similar role as Lemma 2. We caution that the equivalence class returned by this algorithm may be gigantic; it is for this reason that we have advocated the use of some razor (in particular, minimality) to reduce this to a more wieldy set.
> > > > > > > > > > > > >
> > > > > > > > > > > > > For the second case, suppose the adjacency matrix is minimal but not identifiable (we recall from lines 106-110 that every identifiable system is minimal, but not every minimal system is identifiable). In this case, our conditions do not force us to select any system from the equivalence class; this is supported by our simulation study, where every system from the equivalence class fit the data equally well (see lines 287-288 of the supplement).
> > > > > > > > > > > > >
> > > > > > > > > > > > > We conjecture that the equivalence class of observationally equivalent minimal graphs can be generated from a single member by repeated/recursive application of the following three transformations. (Please note this is still just a conjecture. Due to Theorem 2, it is clear that these transformations generate a subset of the equivalence class; the conjecture is that the subset is improper, so that these are the only transformations needed to generate the full equivalence class.) These transformations are lifted directly from the proof of Theorem 2, which also contains details on how the causal weights should be updated.
> > > > > > > > > > > > >
> > > > > > > > > > > > > - If $V_i$ is a weak bottleneck violation, choose an alternative minimal bottleneck $B \neq Ch(V_i)$. Remove every edge from $V_i$ to $Ch(V_i)$, then draw edges from $V_i$ to $B$. Following a similar proof strategy to Proposition 6 in the supplement, it can be shown that there is some $L_j$ such that $Ch(L_j) \subseteq B$ and $L_j \not\to V_i$, so that $L_j$ is now a co-parental redundancy of $V_i$.
> > > > > > > > > > > > >
> > > > > > > > > > > > > - If $L_i$ is a co-parental redundancy of $V_j$, then add $V_j \to L_i$ and remove $V_j \to V_k$ for any single $V_k \in Ch(V_j)$. Note that in the resulting graph, the original $Ch(V_j)$ remains a bottleneck from $V_j$ to $X$, so that $V_j$ now violates the bottleneck condition weakly.
> > > > > > > > > > > > >
> > > > > > > > > > > > > - If $L_i$ is a parental redundancy of $V_j$, then put $Pa’(L_i) = Pa(V_j) – \{L_i\}$, and put $Pa’(V_j) = \{L_i\} \cup Pa(L_i)$. Since $Ch(L_i)$ and $Ch(V_j)$ remain unchanged, $L_i$ remains a parental redundancy of $V_j$. (It is possible in this case that the DAG remains unchanged; we address this in lines 110-114 of the supplement.)
> > > > > > > > > > > > >
> > > > > > > > > > > > > In summary, there is a symmetry between weak bottleneck violations and non-parental redundancies. There is also a symmetry with parental redundancies themselves. For examples of these equivalence classes, see graphs (i-ix) in Figure 6 of the supplement.
> > > > > > > > > > > > >
> > > > > > > > > > > > > Once again, we appreciate your excellent questions and careful review of our paper!

---

> > > > > > > > > > > > > > ### Comment · Reviewer_NMAx · 2021-09-03
> > > > > > > > > > > > > > **Thanks. Let me clarify my understanding**
> > > > > > > > > > > > > >
> > > > > > > > > > > > > > I understood it as the following but is this correct?
> > > > > > > > > > > > > >
> > > > > > > > > > > > > > Suppose that the underlying causal graph is non-minimal, but the other conditions hold. Apply your method on the data. Then, you obtain a minimal causal graph. The two causal graphs are observationally equivalent. Do the two causal graphs share some causal structure, e.g., some topological causal orders of observed variables?
> > > > > > > > > > > > > >
> > > > > > > > > > > > > > In theory, the answers to this question are unknown. But, in practice, well, though trying all the permutations would not be feasible, we could enumerate all the observationally equivalent causal graphs and see which structure is common to the observationally equivalent causal graphs. The observationally equivalent causal graphs include both minimal causal graphs and non-minimal causal graphs.
> > > > > > > > > > > > > >
> > > > > > > > > > > > > > We need some assumptions to identify the causal graph or the topological causal orders of observed variables uniquely. Previous works found some conditions for this. You find another set of conditions that if we accept the minimality condition and other conditions, we can uniquely identify the causal graph. You also showed that the conditions are sufficient and necessary if the minimality condition is assumed to be true. The relations of models satisfying your conditions with those satisfying previous identifiable conditions are not yet unknown.

---

> > > > > > > > > > > > > > > ### Author Response · Authors · 2021-09-03
> > > > > > > > > > > > > > > **Some clarifications of your summary**
> > > > > > > > > > > > > > >
> > > > > > > > > > > > > > > Thank you for the chance to summarize this stimulating discussion! Please find our replies in-line below.
> > > > > > > > > > > > > > >
> > > > > > > > > > > > > > > Q1. “Suppose that the underlying causal graph is non-minimal, but the other conditions hold.”
> > > > > > > > > > > > > > >
> > > > > > > > > > > > > > > A1. By “the other conditions,” we assume you are referring to the linearity condition and the heterogeneity or non-Gaussianity conditions on the noise terms. If the system is not minimal (line 107), then the graphical conditions must fail to hold due to Theorem 3. In other words, minimality is not an additional assumption on top of Conditions 1-3; it is rather entailed by them. For more on this, see A6 and A7 below.
> > > > > > > > > > > > > > >
> > > > > > > > > > > > > > > Q2. “Apply your method on the data. Then, you obtain a minimal causal graph. The two causal graphs are observationally equivalent.”
> > > > > > > > > > > > > > >
> > > > > > > > > > > > > > > A2. We agree; our experiments in Section 2.2 of the supplement support this.
> > > > > > > > > > > > > > >
> > > > > > > > > > > > > > > Q3. “Do the two causal graphs share some causal structure, e.g., some topological causal orders of observed variables? In theory, the answers to this question are unknown.”
> > > > > > > > > > > > > > >
> > > > > > > > > > > > > > > A3. Actually, the answer to your particular question (causal order of observed variables) is “Yes, the observed variables have the same topological order on both graphs as long as the true graph satisfies faithfulness.” (Section 1.3 of the supplement tells us that bottleneck faithfulness, and hence classical faithfulness, is practically never violated in nature.) We know this because Theorem 1 of SGKZ shows that the causal order of observed variables is identifiable in any system that satisfies classical faithfulness, and our model assumptions entail classical faithfulness.
> > > > > > > > > > > > > > >
> > > > > > > > > > > > > > > Q4. “But, in practice, well, though trying all the permutations would not be feasible, we could enumerate all the observationally equivalent causal graphs and see which structure is common to the observationally equivalent causal graphs. The observationally equivalent causal graphs include both minimal causal graphs and non-minimal causal graphs.”
> > > > > > > > > > > > > > >
> > > > > > > > > > > > > > > A4. We agree that enumerating the full class of observationally equivalent graphs is not feasible. It’s just too big. The graphs in this class are also difficult to describe succinctly; we hope our previous response and Figure 6 of the supplement gives some intuition about why this is hard, and that they show what sorts of alternative graphs will appear in the non-minimal equivalence class.
> > > > > > > > > > > > > > >
> > > > > > > > > > > > > > > We agree that finding graphical features that are invariant within this class (in addition to the causal order of observed variables, which we’ve already solved in A3 above) could make for interesting future work.
> > > > > > > > > > > > > > >
> > > > > > > > > > > > > > > With that said, the class of observationally equivalent minimal graphs is usually much smaller (indeed it is a singleton if Conditions 1-3 of the paper are satisfied). We suspect that this minimal equivalence class can be enumerated in reasonable time complexity by leveraging the three symmetries of our previous response.
> > > > > > > > > > > > > > >
> > > > > > > > > > > > > > > Q5. “We need some assumptions to identify the causal graph or the topological causal orders of observed variables uniquely. Previous works found some conditions for this. You find another set of conditions...”
> > > > > > > > > > > > > > >
> > > > > > > > > > > > > > > A5. We completely agree that some assumptions are needed! However, we want to push back on the idea that we have merely found another set of conditions. Every work we have cited takes some form of sparsity as a starting principle (knowingly or unknowingly). Which form is chosen is dictated by the task (for instance, there is no reason to posit minimality on the latent part of the graph unless you’re trying to estimate latent structures). Each previous work then provides some graphical conditions (in our case, the bottleneck and strong non-redundancy conditions) which entail the relevant type of sparsity.
> > > > > > > > > > > > > > >
> > > > > > > > > > > > > > > In our paper, we have taken the version of sparsity which is standard for our task; we have argued previously that references [19-23] implicitly take this same sparsity principle as a starting point, in the sense that their conditions entail it. What is special about our work is that our conditions not only entail this standard sparsity principle, but are also entailed by it. In other words, our conditions cannot be improved upon unless a different starting principle is chosen.
> > > > > > > > > > > > > > >
> > > > > > > > > > > > > > > Q6. “… that if we accept the minimality condition and other conditions, we can uniquely identify the causal graph.”
> > > > > > > > > > > > > > >
> > > > > > > > > > > > > > > A6. We do not require minimality as an additional condition on top of Conditions 1-3; Conditions 1-3 entail minimality (and further, the stronger notion of identifiability up to trivialities) due to Theorem 3.
> > > > > > > > > > > > > > >
> > > > > > > > > > > > > > > Q7. “You also showed that the conditions are sufficient and necessary if the minimality condition is assumed to be true.”
> > > > > > > > > > > > > > >
> > > > > > > > > > > > > > > A7. More precisely, we have shown that a graph is minimal and its causal weights are uniquely determined up to rescaling and reindexing of latent variables if and only if the bottleneck and non-redundancy conditions are satisfied. (The “if” direction also requires bottleneck faithfulness, but Section 1.3 of the supplement basically gives us that for free.)
> > > > > > > > > > > > > > >
> > > > > > > > > > > > > > > Q8. “The relations of models satisfying your conditions with those satisfying previous identifiable conditions are not yet unknown.”
> > > > > > > > > > > > > > >
> > > > > > > > > > > > > > > A8. We believe that the relation of our conditions to previous conditions is fairly well understood. The reply from August 23 discusses this relation in detail. The reply from August 25 further complements that discussion.

---

> > > > > > > > > > > > > > > > ### Comment · Reviewer_NMAx · 2021-09-03
> > > > > > > > > > > > > > > > **Thanks a lot for your feedback!**
> > > > > > > > > > > > > > > >
> > > > > > > > > > > > > > > > Ok, thanks a lot for your feedback!

---

> > > > > > > > > > > > > > > > > ### Comment · Reviewer_NMAx · 2021-09-03
> > > > > > > > > > > > > > > > > **ps**
> > > > > > > > > > > > > > > > >
> > > > > > > > > > > > > > > > > I found that A3 was what I wanted to know by asking those questions.

---

### Official Review · Reviewer_8tp4 · 2021-07-19

**Rating:** 7
**Confidence:** 3

**Summary:**

This paper provides two graphical conditions for identifying a linear causal model under non-Gaussian error or heterogenous errors across domains when a subset of variables are latent. Specifically, the paper introduces necessary (‘unique minimal bottleneck’ criterion and strong non-redundancy) and sufficient conditions (bottleneck faithfulness) for the identifiability. These conditions indeed form an identifiability condition.

**Ethical Concerns:**

.

**Limitations And Societal Impact:**

.

**Main Review:**


originality:
The paper is original. The related work section is comprehensive enough to appreciate the original contributions of this paper.


quality:
Considering the readability, I was pleased to read the paper. Notations, texts, theorems, and explanations are all placed perfectly. Sections are organized coherently. Figures are simple yet informative. Technically, the authors made progress in identifiability via graphical and matrix-related understanding, connecting other existing conditions (as demonstrated in the related work section).


clarity:
Is theorem 1 if and only if condition? It is currently written like sufficiency. Given that some of the space is left, it would be helpful if Section 4.2 contains figures that can explain the Lemmas in the section. Similarly, the section may have an algorithm (pseudocode?).


significance:
This paper provides necessity, sufficiency, and identification results (in addition to estimation). Since (theoretical) causality research focuses on the existence of clearly defined estimands, such conditions will be valuable resources to develop the causal theory further.


limitation:
Although linearity is a strong assumption over the functions of causal models, since it allows us to study a class of causal models in depth, I would not consider the assumption restrictive. In the future, it would be desirable to provide some details about the estimation procedure for overcomplete ICA.


**Time Spent Reviewing:**

6

---

> ### Author Response · Authors · 2021-08-10
> **Thank you for your encouraging comments!**
>
> Thank you for your encouraging comments! We are especially happy you agree that our identifiability conditions will facilitate future research on the problem. For example, we hope that our detailed analysis of the linear case will provide useful insights for future analysis of the non-linear case. Similarly, we believe our conditions will be especially useful in the development of estimation algorithms for partially observed linear systems. In existing work on this topic, quite a lot of effort is spent coming up with tractable special cases of identifiable partially observed causal structures, then proving such structures’ identifiability, and finally developing an efficient estimation algorithm for such structures. We hope that our graphical conditions will streamline the this process in future research, allowing future works to skip straight to the estimation algorithm.
>
> Minor points: Thanks for pointing out that Theorem 1 is a sufficiency result, and not an if-and-only-if. Adding a worked example of Section 4.2 is an excellent idea; we will add such an example in the manuscript.

---

> > ### Comment · Reviewer_8tp4 · 2021-09-01
> > **follow-up**
> >
> > There has been helpful discussions lead by other reviewers. I will maintain the positive assessment.

---

### Official Review · Reviewer_jP5B · 2021-07-20

**Rating:** 7
**Confidence:** 4

**Summary:**

The paper provides necessary and sufficient graphical conditions for full identification of linear causal models with latent variables. It introduces the so-called ‘bottleneck’ and ‘strong-redundancy’ condition in combination with bottleneck faithfulness that together ensure identifiability (up to irrelevant permutations/scaling). It also considers likelihood approaches to estimating the unique solution from data that exploit heterogeneity in the independent noise terms over multiple domains. Experimental results are reported to illustrate feasibility of the method.

**Ethical Concerns:**

no ethical concerns

**Limitations And Societal Impact:**

no negative societal impact

**Main Review:**

Challenging and relevant problem that has been studied in many versions before. The main contribution of this paper is that it provides some generic graphical criteria that determine whether or not this identification problem is uniquely solvable, thereby bringing together several other approaches.

Drawback is that it is hard to asses to what extent the accompanying assumptions apply in practice or whether they are very restrictive or not. Naturally ‘linear’ is often assumed but not always valid. Also the need to know the number of latents beforehand makes it harder to apply in real-world scenarios. Often the goal is to infer what latent variables / constructs can be recognised/identified  from the observed distribution. Similarly, on applying the reconstruction estimation described in section 6, it is not easy to verify from the output obtained whether the assumptions actually did hold or not. And actually determining the full model F itself may involve an exhaustive search over graphs, or hope that a penalised log-likelihood approach will converge to the true optimum quickly … which is by no means guaranteed either.
Perhaps the ‘full identifiability’ is more than what is actually needed. For example, a model with subpath Li -> Lj -> Xk would not satisfy the conditions, but the marginal over Li -> Xk would, and still be equally informative or useful to have. And for the reconstruction problem based on available X that would be perfectly fine.

Technical quality is good, with attention to detail and good efforts to interpret the sometimes quite abstract theoretical results. Clarity can be improved at some points where concepts are used before they are actually defined, but overall very readable for a challenging subject like this. Experimental evaluation could be a bit more extensive but is ok to illustrate some of the remaining challenges.

originality: well-known problem, but interesting and original approach
quality: high quality
clarity: ok with minor suggestions for improvement below
significance: not directly applicable as an off-the-shelf algorithm, but certainly a meaningful theoretical contribution that provides new insights

minor comments:
p3.82: typo ‘the the’
p3.93: ‘uncorrelated in each domain’ => does this mean they all *have* to change between domains? if so, then that is a rather strong and unrealistic setting, as in most cases only a subset of noise terms is likely to differ between domains / over time
p3.116: can the nodes in J and K themselves be part of the bottleneck from J to K? (ok later becomes clear)
p4,123-8: These two assumptions are at first sight quite artificial and hard to parse in terms of what graphs are excluded and what not. In other words, it is not easy to see how restrictive / realistic these assumptions are in practice. It would be helpful to give some hints/pointers to see what types of graphs generally satisfy these conditions. For example if the variables can be separated so that there are no directed paths from X to L, does the causal structure between the X influence the ‘unique minimal bottleneck condition’?
p4.133: explain/define ‘bottleneck faithful’ … (ok later found at p5.156, just refer here)
p6.192: weird character ‘one?’ in the pdf (or perhaps just my reader)
p6.213: ‘only recover limited information about the causal structure’ => true, but it would be interesting to see if starting from these to reduce the full search space to ‘local latent structure discovery’ would make the assumptions provided here much less restrictive in practice … and with it greatly enhance the applicability
p9.321-324: doesn’t L1 typically induce sparsity by driving certain small coefficients to zero?


**Time Spent Reviewing:**

3

---

> ### Author Response · Authors · 2021-08-10
> **Thank you for your time and thoughtful comments!**
>
> We are very grateful to the reviewer for the thoughtful comments and the time devoted to the paper. We have collected them into five main remarks, and addressed them below.
>
> Q1: “Drawback is that it is hard to asses to what extent the accompanying assumptions apply in practice or whether they are very restrictive or not. Naturally ‘linear’ is often assumed but not always valid. Also the need to know the number of latents beforehand makes it harder to apply in real-world scenarios. Often the goal is to infer what latent variables / constructs can be recognised/identified from the observed distribution. Similarly, on applying the reconstruction estimation described in section 6, it is not easy to verify from the output obtained whether the assumptions actually did hold or not.”
>
> A1: Thank you for this concern. We agree that linearity is often assumed but not always valid. The estimation of the number of latents can be cast as a model selection process. Regarding verifying from the output whether the assumptions actually hold, first, non-redundancy is actually local, and quite simple to check --  to check whether it holds between two variables $V_1$ and $V_2$, we only need to know $Ch(V_1)$ and $Ch(V_2)$. On the other hand, the bottleneck condition is relatively global. To check whether the bottleneck condition holds for $L_i$, we must inspect all directed paths from $L_i$ to X, because a minimal bottleneck may lurk arbitrarily far downstream from $L_i$ without compromising the identification of the rest of the graph. We will add an example illustrating this point to the manuscript.
>
> Q2: “These two assumptions are at first sight quite artificial and hard to parse in terms of what graphs are excluded and what not. In other words, it is not easy to see how restrictive / realistic these assumptions are in practice. It would be helpful to give some hints/pointers to see what types of graphs generally satisfy these conditions. For example if the variables can be separated so that there are no directed paths from X to L, does the causal structure between the X influence the ‘unique minimal bottleneck condition’?”
>
> A2: Regarding intuition about the graphical conditions, Figures 1-3 present some simple motivations for the conditions, and Figures 5-6 of the supplement provide some simple examples of small graphs which are respectively identifiable and unidentifiable. Here, let us build intuition by providing some additional consequences of the conditions. For example: each variable must have fewer than $|X|$ children; each latent variable must have at least two children; if a variable has no latent children, then the bottleneck condition is satisfied trivially. We will include these in our manuscript. To answer your particular question, if there are no paths from X to L, then the causal graph among X has no impact on the bottleneck condition. (Proof: if B is a bottleneck from $Ch(V_i)$ to X, then on any path originating in $Ch(V_i)$ and terminating in X, the path must be blocked by B upon or before crossing the first observed node on the path. This fact is independent of that observed variable’s children. Since this holds for any bottleneck, it holds in particular for the minimal bottleneck.) On the other hand, the causal graph among X is relevant for non-redundancy. To see why, please refer to Figure 6(iv-vi) of the supplement, and compare this to Figure 5(viii-xi) of the supplement.
>
> Q3: “Perhaps the ‘full identifiability’ is more than what is actually needed”.
>
> A3: This is a very good point, and it suggests two separate directions for future research. One is to leverage the identification theory of the present paper to find local causal structures which leave easily identified “causal signatures” in the observed data, and which are themselves identifiable even if the graph taken as a whole is not. This is the direction taken by various existing works (including the classical Tetrad conditions). A second line of research could investigate identifiable “versions” of non-identifiable graphs. Such work would extend the “canonical models” in Hoyer et al. [17], but informed by our necessary and sufficient conditions, could consider more sophisticated indeterminacies. By characterizing the relationship between unidentifiable and identifiable structures, this would expand the kinds of partially observed causal systems which can be approximately estimated and studied.
>
> Q4: “p9.321-324: doesn’t L1 typically induce sparsity by driving certain small coefficients to zero?”
>
> A4: Thanks for this comment; we completely agree that L1 typically drives small coefficients to zero. Indeed, in our experiments, L1 does drive some small coefficients to zero. However, our objective function is also highly non-convex, which makes this behavior’s relation to the penalization constant less predictable than in, say, linear regression. Thus we may find local minima that are slightly denser than the true system, but whose every non-zero entry is “medium” sized: too large to prune, but still adding up to a relatively small total L1 penalty. This is discussed in further detail, together with an example, in the supplement (Section 2.3.).
>
> Q5: “p3.93: ‘uncorrelated in each domain’ => does this mean they all have to change between domains? if so, then that is a rather strong and unrealistic setting, as in most cases only a subset of noise terms is likely to differ between domains”
>
> A5: Thanks for raising this excellent point. In order to establish full identifiability of the underlying structure in the paper, we assume that the variances of all noise terms change between domains or over time. However, we completely agree that in practice perhaps only a subset of noise terms will ever change, with the rest staying constant. Establishing partial identifiability in this setting will be very interesting and extremely useful in practice, although it does not seem straightforward. We believe it will be a meaningful direction of future research.
>
> In addition, thank you for catching various minor typos. We have corrected them in the manuscript.

---

### Decision · Program_Chairs · 2021-09-27

**Decision:**

Accept (Poster)

**Comment:**

The reviewers appreciated the relevance of the paper to the NeurIPS community and the strength of the technical results.

Additionally, the authors did a good job of engaging with the reviewers to dispel most concerns. Hopefully, the discussion clarified to the authors which parts of the paper are less readable, and they can use this information to improve the writing.